# TRAINING BATCHNORM AND ONLY BATCHNORM: ON THE EXPRESSIVE POWER OF RANDOM FEATURES IN CNNS

**Jonathan Frankle**[*]
MIT CSAIL
jfrankle@mit.edu

**David J. Schwab**
CUNY Graduate Center, ITS
Facebook AI Research
dschwab@fb.com

**Ari S. Morcos**
Facebook AI Research
arimorcos@fb.com

## ABSTRACT

A wide variety of deep learning techniques from style transfer to multitask learning rely on training affine transformations of features. Most prominent among these is the popular feature normalization technique BatchNorm, which normalizes activations and then subsequently applies a learned affine transform. In this paper, we aim to understand the role and expressive power of affine parameters used to transform features in this way. To isolate the contribution of these parameters from that of the learned features they transform, we investigate the performance achieved when training *only* these parameters in BatchNorm and freezing all weights at their random initializations. Doing so leads to surprisingly high performance considering the significant limitations that this style of training imposes. For example, sufficiently deep ResNets reach 82% (CIFAR-10) and 32% (ImageNet, top-5) accuracy in this configuration, far higher than when training an equivalent number of randomly chosen parameters elsewhere in the network. BatchNorm achieves this performance in part by naturally learning to disable around a third of the random features. Not only do these results highlight the expressive power of affine parameters in deep learning, but—in a broader sense—they characterize the expressive power of neural networks constructed simply by shifting and rescaling random features.

## 1 INTRODUCTION

Throughout the literature on deep learning, a wide variety of techniques rely on learning affine transformations of features—multiplying each feature by a learned coefficient $\gamma$ and adding a learned bias $\beta$. This includes everything from multi-task learning (Mudrakarta et al., 2019) to style transfer and generation (e.g., Dumoulin et al., 2017; Huang & Belongie, 2017; Karras et al., 2019). One of the most common examples of these affine parameters are in feature normalization techniques like BatchNorm (Ioffe & Szegedy, 2015). Considering their practical importance and their presence in nearly all modern neural networks, we know relatively little about the role and expressive power of affine parameters used to transform features in this way.

To gain insight into this question, we focus on the $\gamma$ and $\beta$ parameters in BatchNorm. BatchNorm is nearly ubiquitous in deep convolutional neural networks (CNNs) for computer vision, meaning these affine parameters are present by default in numerous models that researchers and practitioners train every day. Computing BatchNorm proceeds in two steps during training (see Appendix A for full details). First, each pre-activation[1] is normalized according to the mean and standard deviation across the mini-batch. These normalized pre-activations are then scaled and shifted by a trainable per-feature coefficient $\gamma$ and bias $\beta$.

One fact we do know about $\gamma$ and $\beta$ in BatchNorm is that their presence has a meaningful effect on the performance of ResNets, improving accuracy by 0.5% to 2% on CIFAR-10 (Krizhevsky et al., 2009) and 2% on ImageNet (Deng et al., 2009) (Figure 1). These improvements are large enough

---

[*]Work done while an intern and student researcher at Facebook AI Research.
[1]He et al. (2016) find better accuracy when using BatchNorm before activation rather than after in ResNets.

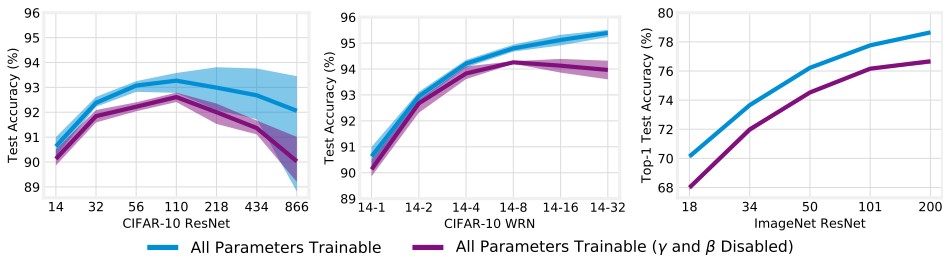

Figure 1: Accuracy when training deep (left) and wide (center) ResNets for CIFAR-10 and deep ResNets for ImageNet (right) as described in Table 1 when all parameters are trainable (blue) and all parameters except $\gamma$ and $\beta$ are trainable (purple). Training with $\gamma$ and $\beta$ enabled results in accuracy 0.5% to 2% (CIFAR-10) and 2% (ImageNet) higher than with $\gamma$ and $\beta$ disabled.

that, were $\gamma$ and $\beta$ proposed as a new technique, it would likely see wide adoption. However, they are small enough that it is difficult to isolate the specific role $\gamma$ and $\beta$ play in these improvements.

More generally, the central challenge of scientifically investigating per-feature affine parameters is distinguishing their contribution from that of the features they transform. In all practical contexts, these affine parameters are trained jointly with (as in the case of BatchNorm) or after the features themselves (Mudrakarta et al., 2019; Dumoulin et al., 2017). In order to study these parameters in isolation, we instead train them on a network composed entirely of random features. Concretely, we freeze all weights at initialization and train *only* the $\gamma$ and $\beta$ parameters in BatchNorm.

Although the networks still retain the same number of features, only a small fraction of parameters (at most 0.6%) are trainable. This experiment forces all learning to take place in $\gamma$ and $\beta$, making it possible to assess the expressive power of a network whose only degree of freedom is scaling and shifting random features. We emphasize that our goal is scientific in nature: to assess the performance and the mechanisms by which networks use this limited capacity to represent meaningful functions; we neither intend nor expect this experiment to reach SOTA accuracy. We make the following findings:

- When training only $\gamma$ and $\beta$, sufficiently deep networks (e.g., ResNet-866 and ResNet-200) reach surprisingly high (although non-SOTA) accuracy: 82% on CIFAR-10 and 32% top-5 on ImageNet. This demonstrates the expressive power of the affine BatchNorm parameters.
- Training an equivalent number of randomly-selected parameters per channel performs far worse (56% on CIFAR-10 and 4% top-5 on ImageNet). This demonstrates that $\gamma$ and $\beta$ have particularly significant expressive power as per-feature coefficients and biases.
- When training only BatchNorm, $\gamma$ naturally learns to disable between a quarter to half of all channels by converging to values close to zero. This demonstrates that $\gamma$ and $\beta$ achieve this accuracy in part by imposing per-feature sparsity.
- When training all parameters, deeper and wider networks have smaller $\gamma$ values but few features are outright disabled. This hints at the role $\gamma$ may play in moderating activations in settings where disabling $\gamma$ and $\beta$ leads to lower accuracy (the right parts of the plots in Figure 1).

In summary, we find that $\gamma$ and $\beta$ have noteworthy expressive power in their own right and that this expressive power results from their particular position as a per-feature coefficient and bias. Beyond offering insights into affine parameters that transform features, this observation has broader implications for our understanding of neural networks composed of random features. By freezing all other parameters at initialization, we are training networks constructed by learning shifts and rescalings of random features. In this light, our results demonstrate that the random features available at initialization provide sufficient raw material to represent high-accuracy functions for image classification. Although prior work considers models with random features and a trainable linear output layer (e.g., Rahimi & Recht, 2009; Jaeger, 2003; Maass et al., 2002), we reveal the expressive power of networks configured such that trainable affine parameters appear after each random feature.

## 2 RELATED WORK

**BatchNorm.** BatchNorm makes it possible to train deeper networks (He et al., 2015a) and causes SGD to converge sooner (Ioffe & Szegedy, 2015). However, the underlying mechanisms by which it

does so are debated. The original authors argue it reduces *internal covariate shift* (ICS), in which "the distribution of each layer's inputs changes during training...requiring lower learning rates" (Ioffe & Szegedy, 2015). Santurkar et al. (2018) cast doubt on this explanation by artificially inducing ICS after BatchNorm with little change in training times. Empirical evidence suggests BatchNorm makes the optimization landscape smoother (Santurkar et al., 2018); is a "safety precaution" against exploding activations that lead to divergence (Bjorck et al., 2018); and allows the network to better utilize neurons (Balduzzi et al., 2017; Morcos et al., 2018). Theoretical results suggest Batch-Norm decouples optimization of weight magnitude and direction (Kohler et al., 2019) as *weight normalization* (Salimans & Kingma, 2016) does explicitly; that it causes gradient magnitudes to reach equilibrium (Yang et al., 2019); and that it leads to a novel form of regularization (Luo et al., 2019).

We focus on the role and expressive power of the affine parameters in particular, whereas the aforementioned work addresses the overall effect of BatchNorm on the optimization process. In service of this broader goal, related work generally emphasizes the normalization aspect of Batch-Norm, in some cases eliding one (Kohler et al., 2019) or both of $\gamma$ and $\beta$ (Santurkar et al., 2018; Yang et al., 2019). Other work treats BatchNorm as a black-box without specific consideration for $\gamma$ and $\beta$ (Santurkar et al., 2018; Bjorck et al., 2018; Morcos et al., 2018; Balduzzi et al., 2017). One notable exception is the work of Luo et al. (2019), who show theoretically that BatchNorm imposes a $\gamma$ *decay*—a data-dependent L2 penalty on $\gamma$; we discuss this work further in Section 5.

**Exploiting the expressive power of affine transformations.** There are many techniques in the deep learning literature that exploit the expressive power of affine transformations of features. Mudrakarta et al. (2019) develop a parameter-efficient approach to multi-task learning with a shared network backbone (trained on a particular task) and separate sets of per-task BatchNorm parameters (trained on a different task while the backbone is frozen). Similarly, Rebuffi et al. (2017) allow a network backbone trained on one task to adapt to others by way of residual models added onto the network comprising BatchNorm and a convolution. Perez et al. (2017) perform visual question-answering by using a RNN receiving text input to produce coefficients and biases that are used to transform the internal features of a CNN on visual data. Finally, work on neural style transfer and style generation uses affine transformations of normalized features to encode different styles (e.g., Dumoulin et al., 2017; Huang & Belongie, 2017; Karras et al., 2019).

**Training only BatchNorm.** Closest to our work, Rosenfeld & Tsotsos (2019) explore freezing various parts of networks at initialization; in doing so, they briefly examine training only $\gamma$ and $\beta$. However, there are several important distinctions between this paper and our work. They conclude only that it is generally possible to "successfully train[] mostly-random networks," while we find that BatchNorm parameters have greater expressive power than other parameters (Figure 2, green).

In fact, their experiments cannot make this distinction. They train only BatchNorm in just two CIFAR-10 networks (DenseNet and an unspecified Wide ResNet) for just ten epochs (vs. the standard 100+), reaching 61% and 30% accuracy. For comparable parameter-counts, we reach 80% and 70%. These differences meaningfully affect our conclusions: they allow us to determine that training only BatchNorm leads to demonstrably higher accuracy than training an equivalent number of randomly chosen parameters. The accuracy in Rosenfeld & Tsotsos is too low to make any such distinction.

Moreover, we go much further in terms of both scale of experiments and depth of analysis. We study a much wider range of networks and, critically, show that training only BatchNorm can achieve impressive results even for large-scale networks on ImageNet. We also investigate *how* the BatchNorm parameters achieve this performance by examining the underlying representations.

We also note that Mudrakarta et al. (2019) train only BatchNorm and a linear output layer on a single, randomly initialized MobileNet (in the context of doing so on many trained networks for the purpose of multi-task learning); they conclude simply that "it is quite striking" that this configuration "can achieve non-trivial accuracy."

**Random features.** There is a long history of building models from random features. The perceptron (Block, 1962) learns a linear combination of *associators*, each the inner product of the input and a random vector. More recently, Rahimi & Recht (2009) showed theoretically and empirically that linear combinations of random features perform nearly as well as then-standard SVMs and Adaboost. *Reservoir computing* (Schrauwen et al., 2007), also known as *echo state networks* (Jaeger, 2003) or *liquid state machines* (Maass et al., 2002), learns a linear readout from a randomly connected recurrent neural network; such models can learn useful functions of sequential data. To theoretically

| Family | ResNet for CIFAR-10 | | | | | | | Wide ResNet (WRN) for CIFAR-10 | | | | | | ResNet for ImageNet | | | | |
|---|---|---|---|---|---|---|---|---|---|---|---|---|---|---|---|---|---|---|
| Depth | 14 | 32 | 56 | 110 | 218 | 434 | 866 | 14 | 14 | 14 | 14 | 14 | 14 | 18 | 34 | 50 | 101 | 200 |
| Width Scale | 1 | 1 | 1 | 1 | 1 | 1 | 1 | 1 | 2 | 4 | 8 | 16 | 32 | 1 | 1 | 1 | 1 | 1 |
| Total | 175K | 467K | 856K | 1.73M | 3.48M | 6.98M | 14.0M | 175K | 696K | 2.78M | 11.1M | 44.3M | 177M | 11.7M | 21.8M | 25.6M | 44.6M | 64.7M |
| BatchNorm | 1.12K | 2.46K | 4.26K | 8.29K | 16.4K | 32.5K | 64.7K | 1.12K | 2.24K | 4.48K | 8.96K | 17.9K | 35.8K | 9.6K | 17.0K | 53.1K | 105K | 176K |
| Output | 650 | 650 | 650 | 650 | 650 | 650 | 650 | 650 | 1.29K | 2.57K | 5.13K | 10.3K | 20.5K | 513K | 513K | 2.05M | 2.05M | 2.05M |
| Shortcut | 2.56K | 2.56K | 2.56K | 2.56K | 2.56K | 2.56K | 2.56K | 2.56K | 10.2K | 41.0K | 164K | 655K | 2.62M | 172K | 172K | 2.77M | 2.77M | 2.77M |
| BatchNorm | 0.64% | 0.53% | 0.50% | 0.48% | 0.47% | 0.47% | 0.46% | 0.64% | 0.32% | 0.16% | 0.08% | 0.04% | 0.02% | 0.08% | 0.08% | 0.21% | 0.24% | 0.27% |
| Output | 0.37% | 0.14% | 0.08% | 0.04% | 0.02% | 0.01% | 0.01% | 0.37% | 0.19% | 0.09% | 0.05% | 0.02% | 0.01% | 4.39% | 2.35% | 8.02% | 4.60% | 3.17% |
| Shortcut | 1.46% | 0.55% | 0.30% | 0.15% | 0.07% | 0.04% | 0.02% | 1.46% | 1.47% | 1.47% | 1.48% | 1.48% | 1.48% | 1.47% | 0.79% | 10.83% | 6.22% | 4.28% |

Table 1: ResNet specifications and parameters in each part of the network. ResNets are called *ResNet-D*, where $D$ is the depth. Wide ResNets are called *WRN-D-W*, where $W$ is the width scale. ResNet-18 and 34 have a different block structure than deeper ImageNet ResNets (He et al., 2015a).

study SGD on overparameterized networks, recent work uses two layer models with the first layer wide enough that it changes little during training (e.g., Du et al., 2019); in the limit, the first layer can be treated as frozen at its random initialization (Jacot et al., 2018; Yehudai & Shamir, 2019).

In all cases, these lines of work study models composed of a trainable linear layer on top of random nonlinear features. In contrast, our models have affine trainable parameters *throughout* the network after each random feature in each layer. Moreover, due to the practice of placing BatchNorm before the activation function (He et al., 2016), our affine parameters occur prior to the nonlinearity.

**Freezing weights at random initialization.** Neural networks are initialized randomly (He et al., 2015b; Glorot & Bengio, 2010), and performance with these weights is no better than chance. However, it is still possible to reach high accuracy while retaining some or all of these weights. Zhang et al. (2019a) show that many individual layers in trained CNNs can be reset to their random i.i.d. initializations with little impact on accuracy. Zhou et al. (2019) and Ramanujan et al. (2019) reach high accuracy on CIFAR-10 and ImageNet merely by learning which individual weights to remove.

## 3 METHODOLOGY

**ResNet architectures.** We train convolutional networks with residual connections (*ResNets*) on CIFAR-10 and ImageNet. We focus on ResNets because they make it possible to add features by arbitrarily (a) increasing depth without interfering with optimization and (b) increasing width without parameter-counts becoming so large that training is infeasible. Training deep ResNets generally requires BatchNorm, so it is a natural setting for our experiments. In Appendix C, we run the same experiments for a non-residual VGG-style network for CIFAR-10, finding qualitatively similar results.

We use the ResNets for CIFAR-10 and ImageNet designed by He et al. (2015a).[2] We scale depth according to He et al. (2015a) and scale width by multiplicatively increasing the channels per layer. As depth increases, networks maintain the same number of shortcut and output parameters, but deeper networks have more features and, therefore, more BatchNorm parameters. As width increases, the number of BatchNorm and output parameters increases linearly, and the number of convolutional and shortcut parameters increase quadratically because the number of incoming and outgoing channels both increase. The architectures we use are summarized in Table 1 (full details in Appendix B).

**BatchNorm.** We place BatchNorm before activation, which He et al. (2016) find leads to better performance than placing it after activation. We initialize $\beta$ to 0 and sample $\gamma$ uniformly between 0 and 1, although we consider other initializations in Appendix E.

**Replicates.** All experiments are shown as the mean across five (CIFAR-10) or three (ImageNet) runs with different initializations, data orders, and augmentation. Error bars for one standard deviation from the mean are present in all plots; in many cases, error bars are too small to be visible.

---

[2]CIFAR-10 and ImageNet ResNets are different architecture families with different widths and block designs.

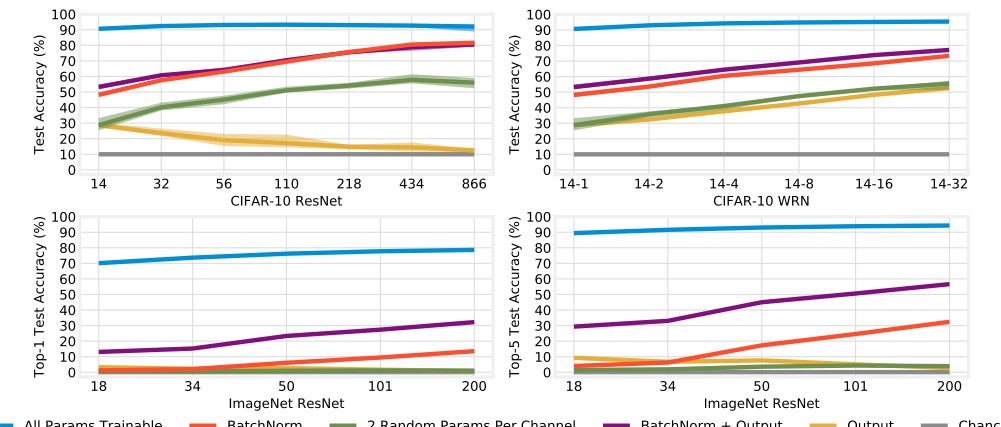

Figure 2: Accuracy of ResNets for CIFAR-10 (top left, deep; top right, wide) and ImageNet (bottom left, top-1 accuracy; bottom right, top-5 accuracy) with different sets of parameters trainable.

# 4   TRAINING ONLY BATCHNORM

In this section, we study freezing all other weights at initialization and train *only* $\gamma$ and $\beta$. These parameters comprise no more than 0.64% of all parameters in networks for CIFAR-10 and 0.27% in networks for ImageNet. Figure 2 shows the accuracy when training only $\gamma$ and $\beta$ in red for the families of ResNets described in Table 1. We also include two baselines: training all parameters (i.e., training normally) in blue and chance performance (i.e., random guessing on the test set) in gray.

**Case study: ResNet-110.** We first consider ResNet-110 on CIFAR-10. When all 1.7M parameters are trainable (blue), the network reaches $93.3\%$ test accuracy. Since CIFAR-10 has ten classes, chance performance is 10%. When training just the 8.3K (0.48%) affine parameters that can only shift and rescale random features, the network achieves surprisingly high test accuracy of 69.5%, suggesting that these parameters have noteworthy representational capacity.

While our motivation is to study the role of the affine parameters, this result also has implications for the expressive power of neural networks composed of random features. All of the features in the network (i.e., convolutions and linear output layer) are fixed at random initializations; the affine parameters can only shift and scale the normalized activation maps that these features produce in each layer. In other words, this experiment can be seen as training neural networks parameterized by shifts and rescalings of random features. In this light, these results show that it is possible to reach high accuracy on CIFAR-10 using only the random features that were available at initialization.

**Increasing available features by varying depth and width.** From the lens of random features, the expressivity of the network will be limited by the number of features available for the affine parameters to combine. If we increase the number of features, we expect that accuracy will improve. We can do so in two ways: increasing the network's depth or increasing its width.

Figure 2 presents the test accuracy when increasing the depth (top left) and width (top right) of CIFAR-10 ResNets and the depth of ImageNet ResNets (bottom). As expected, the accuracy of training only BatchNorm improves as we deepen or widen the network. ResNet-14 achieves 48% accuracy on CIFAR-10 when training only BatchNorm, but deepening the network to 866 layers or widening it by a factor of 32 increases accuracy to 82% and 73%, respectively.Similarly, ResNet-50 achieves 17% top-5 accuracy on ImageNet, but deepening to 200 layers increases accuracy to 32%. [3]

It is possible that, since ImageNet has 1000 classes, accuracy is artificially constrained when freezing the linear output layer because the network cannot learn fine-grained distinctions between classes. To examine this possibility, we made the 0.5M to 2.1M output parameters trainable (Figure 2, purple). Training the output layer alongside BatchNorm improves top-5 accuracy by about 25 percentage points to a maximum value of 57% and top-1 accuracy by 12 to 19 percentage points to a maximum value of 32%. The affine parameters are essential for this performance: training outputs alone

---

[3]In Appendix E, we find that changing the BatchNorm initialization improves accuracy by a further 2-3 percentage points (CIFAR-10) and five percentage points (ImageNet top-5).

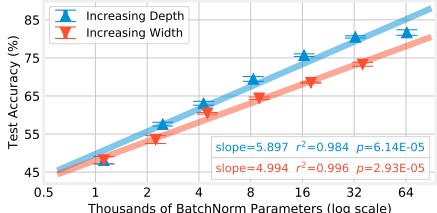

Figure 3: The relationship between BatchNorm parameter count and accuracy when scaling depth and width of CIFAR-10 ResNets.

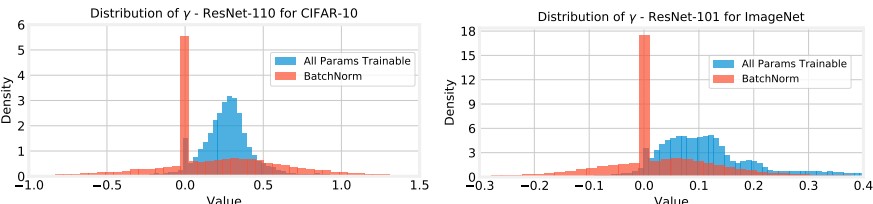

Figure 4: The distribution of $\gamma$ for ResNet-110 and ResNet-101 aggregated from five (CIFAR-10) or three replicates (ImageNet). Distributions of $\gamma$ and $\beta$ for all networks are in Appendix H.

achieves just 2.7% top-5 and 0.8% top-1 accuracy for ResNet-200 (yellow). The same modification makes little difference on CIFAR-10, which has only ten classes.

Finally, note that accuracy is 7 percentage points higher for ResNet-434 than for WRN-14-32 although both have similar numbers of BatchNorm parameters (32.5K vs. 35.8K). This raises a further question: for a fixed budget of BatchNorm parameters (and, thereby, a fixed number of random features), is performance always better when increasing depth rather than increasing width? Figure 3 plots the relationship between number of BatchNorm parameters (x-axis) and test accuracy on CIFAR-10 (y-axis) when increasing depth (blue) and width (red) from the common starting point of ResNet-14. In both cases, accuracy increases linearly as BatchNorm parameter count doubles. The trend is 18% steeper when increasing depth than width, meaning that, for the networks we consider, increasing depth leads to higher accuracy than increasing width for a fixed BatchNorm parameter budget.[4]

**Are affine parameters special?** Is the accuracy of training only BatchNorm a product of the unusual position of $\gamma$ and $\beta$ as scaling and shifting entire features, or is it simply due to the fact that, in aggregate, a substantial number of parameters are still trainable? For example, the 65K BatchNorm parameters in ResNet-866 are a third of the 175K parameters in *all* of ResNet-14; perhaps any arbitrary collection of this many parameters would lead to equally high accuracy.

To assess this possibility, we train two random parameters in each convolutional channel as substitutes for $\gamma$ and $\beta$ (Figure 2, green).[5] Should accuracy match that of training only BatchNorm, it would suggest our observations are not unique to $\gamma$ and $\beta$ and simply describe training an arbitrary subset of parameters as suggested by Rosenfeld & Tsotsos (2019). Instead, accuracy is 17 to 21 percentage points lower on CIFAR-10 and never exceeds 4% top-5 on ImageNet. This result suggests that $\gamma$ and $\beta$ have a greater impact on accuracy than other kinds of parameters.[6] In other words, it appears more important to have coarse-grained control over entire random features than to learn small axis-aligned modifications of the features themselves. [7]

**Summary.** Our goal was to study the role and expressive power of the affine parameters $\gamma$ and $\beta$ in isolation—without the presence of trained features to transform. We found that training only these parameters in ResNets with BatchNorm leads to surprisingly high accuracy (albeit lower than training all parameters). By increasing the quantity of these parameters and the random features they

---

[4]We expect accuracy will eventually saturate and further expansion will have diminishing returns. We begin to see saturation for ResNet-866, which is below the regression line.

[5]We also tried distributing these parameters randomly throughout the layer. Accuracy was the same or lower.

[6]In Appendix F, we find that it is necessary to train between 8 and 16 random parameters per channel on the CIFAR-10 ResNets to match the performance of training only the 2 affine parameters per channel.

[7]In Appendix D, we investigate whether it is better to have a small number of dense trainable features or to learn to scale and shift a large number of random features. To do so, we compare the performance of training only BatchNorm to training all parameters in ResNets with an similar number of trainable parameters.

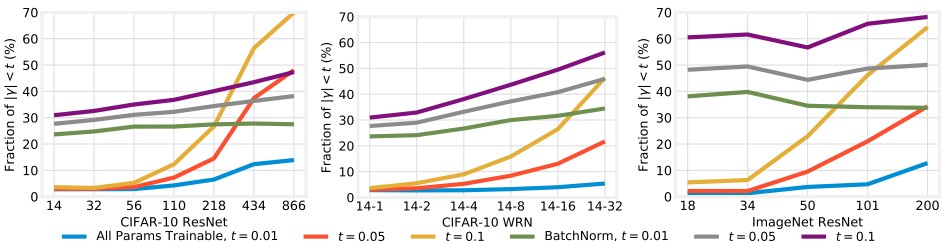

Figure 5: Fraction of $\gamma$ parameters for which $|\gamma|$ is smaller than various thresholds.

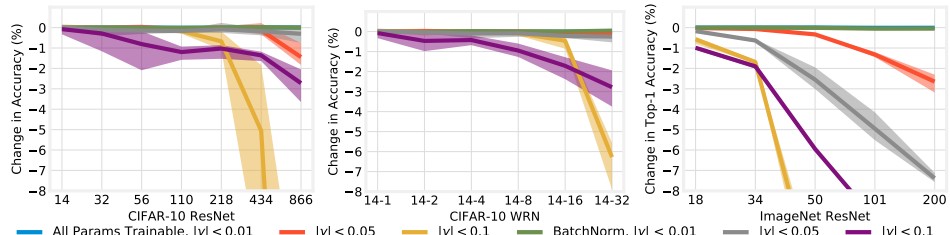

Figure 6: Accuracy change when clamping $\gamma$ values with $|\gamma|$ below various thresholds to 0.

combine, we found that we can further improve this accuracy. This accuracy is not simply due to the raw number of trainable parameters, suggesting that $\gamma$ and $\beta$ have particular expressive power as a per-feature coefficient and bias.

## 5 EXAMINING THE VALUES OF $\gamma$ AND $\beta$

In the previous section, we showed that training just $\gamma$ and $\beta$ leads to surprisingly high accuracy. Considering the severe restrictions placed on the network by freezing all features at their random initializations, we are interested in *how* the network achieves this performance. In what ways do the values and role of $\gamma$ and $\beta$ change between this training regime and when all parameters are trainable?

**Examining $\gamma$.** As an initial case study, we plot the $\gamma$ values learned by ResNet-110 for CIFAR-10 and ResNet-101 for ImageNet when all parameters are trainable (blue) and when only $\gamma$ and $\beta$ are trainable (red) in Figure 4 (distributions for all networks and for $\beta$ are in Appendix H). When training all parameters, the distribution of $\gamma$ for ResNet-110 is roughly normal with a mean of 0.27; the standard deviation of 0.21 is such that 95% of $\gamma$ values are positive. When training only BatchNorm, the distribution of $\gamma$ has a similar mean (0.20) but a much wider standard deviation (0.48), meaning that 25% of $\gamma$ values are negative. For ResNet-101, the mean value of $\gamma$ similarly drops from 0.14 to 0.05 and the standard deviation increases from 0.14 to 0.26 when training only BatchNorm.

Most notably, the BatchNorm-only $\gamma$ values have a spike at 0: 27% (ResNet-110) and 33% (ResNet-101) of all $\gamma$ values have a magnitude $< 0.01$ (compared with 4% and 5% when training all parameters). By setting $\gamma$ so close to zero, the network seemingly learns to disable between a quarter and a third of all features. Other than standard weight decay for these architectures, we take no additional steps to induce this sparsity; it occurs naturally when we train in this fashion. This behavior indicates that an important part of the network's representation is the set of random features that it learns to *ignore*. When all parameters are trainable, there is a much smaller spike at 0, suggesting that disabling features is a natural behavior of $\gamma$, although it is exaggerated when only $\gamma$ and $\beta$ are trainable.

The same behavior holds across all depths and widths, seemingly disabling a large fraction of features. When training only BatchNorm, $|\gamma| < 0.01$ in between a quarter and a third of cases (Figure 5, green). In contrast, when all parameters are trainable (Figure 5, blue), this occurs for just 5% of $\gamma$ values in all but the deepest ResNets. Values of $\gamma$ tend to be smaller for deeper and wider networks, especially when all parameters are trainable. For example, the fraction of $|\gamma| < 0.05$ increases from 3% for ResNet-14 to 48% for ResNet-866. We hypothesize that $\gamma$ values become smaller to prevent exploding activations; this might explain why disabling $\gamma$ and $\beta$ particularly hurts the accuracy of deeper and wider CIFAR-10 networks in Figure 1.

**Small values of $\gamma$ disable features.** Just because values of $\gamma$ are *close* to zero does not necessarily mean they disable features and can be set *equal* to zero; they may still play an important role in the

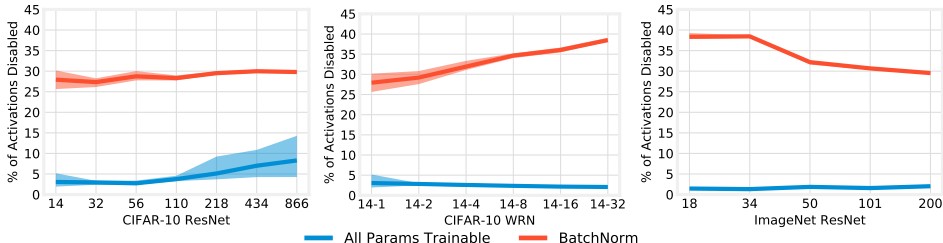

Figure 7: The fraction of ReLU activations for which $\Pr[\text{activation} = 0] > 0.99$.

representation. To evaluate the extent to which small values of $\gamma$ are, in fact, removing features, we explicitly set these parameters to zero and measure the accuracy of the network that results (Figure 6). Clamping all values of $|\gamma| < 0.01$ to zero does not affect accuracy, suggesting that these features are indeed expendable. This is true both when all parameters are trainable and when only BatchNorm is trainable; in the latter case, this means between 24% to 38% of features can be disabled. This confirms our hypothesis that $\gamma$ values closest to zero reflect features that are irrelevant to the network's representation. Results are mixed for a threshold of 0.05: when all parameters are trainable, accuracy remains close to its full value for all but the deepest and widest networks (where we saw a spike in the fraction of parameters below this threshold).

**Training only BatchNorm sparsifies activations.** So far, we have focused solely on the role of $\gamma$. However, $\gamma$ works collaboratively with $\beta$ to change the distribution of normalized pre-activations. It is challenging to describe the effect of $\beta$ due to its additive role; for example, while many $\beta$ values are also close to zero when training only BatchNorm (Appendix H), this does not necessarily disable features. To understand the joint role of $\gamma$ and $\beta$, we study the behavior of the activations themselves. In Figure 7, we plot the fraction of ReLU activations for which $\Pr[\text{activation} = 0] > 0.99$ across all test examples and all pixels in the corresponding activation maps.[8] When training only BatchNorm, 28% to 39% of activations are disabled, meaning $\gamma$ and $\beta$ indeed sparsify activations in practice. In contrast, when all parameters are trainable, no more than 10% (CIFAR-10) and 2% (ImageNet) of activations are disabled according to this heuristic. These results support our hypothesis of the different roles that small values of $\gamma$ play in these two training settings. When only training BatchNorm, we see small values of $\gamma$ and entire activations disabled. However, when all parameters are trainable, few activations are disabled even though a large fraction of $\gamma$ values are small in deeper and wider networks, suggesting that these parameters still play a role in the learned representations.

**Context in the literature.** Mehta et al. (2019) also find that feature-level sparsity emerges in CNNs when trained with certain combinations of optimizers and regularization. They measure this sparsity in a manner similar to ours: the per-feature activations and $\gamma$ values. Mehta et al. only study standard training (not training only BatchNorm). In this context, they find much higher levels of sparsity (sometimes higher than 50%) than we do (less than 5% in all cases we consider), despite the fact that our threshold for $\gamma$ to represent a pruned feature is 10x higher (i.e., we sparsify more features). This may indicate that the behaviors Mehta et al. observe in their "BasicNet" setting (a 7-layer ConvNet for CIFAR) and VGG setting (it appears they use the ImageNet version of VGG, 500x larger than ResNet-20 and very overparameterized for CIFAR-10) may not hold in general.

In their theoretical analysis of BatchNorm, Luo et al. (2019) find that one of its effects is $\gamma$ *decay*—a data-dependent L2 penalty on the $\gamma$ terms in BatchNorm. This observation may provide insight into why we find per-feature sparsity when training only BatchNorm in Section 5: by freezing all other parameters at their initial values, it is possible that our networks behave according to Luo et al.'s assumptions and that $\gamma$ decay may somehow encourage sparsity (although it is not subject to sparsity-inducing L1 regularization).

**Summary.** In this section, we compared the internal representations of the networks when training all parameters and training only the affine parameters $\gamma$ and $\beta$ in BatchNorm. When training only BatchNorm, we found $\gamma$ to have a larger variance and a spike at 0 and that $\gamma$ was learning to disable entire features. When all parameters were trainable, we found that $\gamma$ values became smaller in wider

---

[8]For some features, this is also the fraction of batch-normalized pre-activations that are $\leq 0$ (i.e., that will be eliminated by the ReLU). However, at the end of a residual block, the batch-normalized pre-activations are added to the skip connection before the ReLU, so even if $\gamma = 0$, the activation may be non-zero.

and deeper networks but activations were not disabled, which implies that these parameters still play a role in these networks.

## 6 DISCUSSION AND CONCLUSIONS

Our results demonstrate that it is possible to reach surprisingly high accuracy when training only the affine parameters associated with BatchNorm and freezing all other parameters at their original initializations. To answer our research question, we conclude that affine parameters that transform features have substantial expressive power in their own right, even when they are not paired with learned features. We make several observations about the implications of these results.

**BatchNorm.** Although the research community typically focuses on the normalization aspect of BatchNorm, our results emphasize that the affine parameters are remarkable in their own right. Their presence tangibly improves performance, especially in deeper and wider networks (Figure 1), a behavior we connect to our observation that values of $\gamma$ are smaller as the networks become deeper. On their own, $\gamma$ and $\beta$ create surprisingly high-accuracy networks, even compared to training other subsets of parameters, despite (or perhaps due to) the fact that they disable more than a quarter of activations.

**Random features.** From a different perspective, our experiment is a novel way of training networks constructed out of random features. While prior work (e.g., Rahimi & Recht, 2009) considers training only a linear output layer on top of random nonlinear features, we distribute affine parameters throughout the network after each feature in each layer. This configuration appears to give the network greater expressive power than training the output layer alone (Figure 2). Empirically, we see our results as further evidence (alongside the work of Zhou et al. (2019) and Ramanujan et al. (2019)) that the raw material present at random initialization is sufficient to create performant networks. It would also be interesting to better understand the theoretical capabilities of our configuration.

Unlike Rahimi & Recht, our method does not provide a practical reduction in training costs; it is still necessary to fully backpropagate to update the deep BatchNorm parameters. However, our work could provide opportunities to reduce the cost of storing networks at inference-time. In particular, rather than needing to store all of the parameters in the network, we could store the random seed necessary to generate the network's weights and the trained BatchNorm parameters. We could even do so in a multi-task fashion similar to Mudrakarta et al. (2019), storing a single random seed and multiple sets of BatchNorm parameters for different tasks. Finally, if we are indeed able to develop initialization schemes that produce random features that lead to higher accuracy on specific tasks, we could store a per-task distribution and random seed alongside the per-task BatchNorm parameters.

**Limitations and future work.** There are several ways to expand our study to improve the confidence and generality of our results. We only consider ResNets trained on CIFAR-10 and ImageNet, and it would be valuable to consider other architecture families and tasks (e.g., Inception on computer vision and Transformer on NLP). In addition, we use standard hyperparameters and do not search for hyperparameters that specifically perform well when training only BatchNorm.

In follow-up work, we are interested in further studying the relationship between random features and the representations learned by the affine parameters. Are there initialization schemes for the convolutional layers that allow training only the affine parameters to reach better performance than using conventional initializations? (See Appendix E for initial experiments on this topic.) Is it possible to rejuvenate convolutional filters that are eliminated by $\gamma$ (in a manner similar to Cohen et al. (2016)) to improve the overall accuracy of the network? Finally, can we better understand the role of these affine parameters outside the context of BatchNorm? That is, can we distinguish the expressive power of the affine parameters from the normalization itself? For example, we could add these parameters when using techniques that train deep networks without normalization, such as WeightNorm (Salimans & Kingma, 2016) and FixUp initialization (Zhang et al., 2019b).

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

TABLE OF CONTENTS FOR SUPPLEMENTARY MATERIAL

In these appendices, we include additional details about our experiments, additional data that did not fit in the main body of the paper, and additional experiments. The appendices are as follows:

**Appendix A.** A formal re-statement of the standard BatchNorm algorithm.

**Appendix B.** The details of the ResNet architectures and training hyperparameters we use.

**Appendix C.** The experiments from the main body of this paper performed on the VGG family of architectures for CIFAR-10, which do not have residual connections. The results match those described in the main body.

**Appendix D.** Comparing the performance of training only BatchNorm to training small ResNets with an equivalent number of trainable parameters.

**Appendix E.** The effect of varying the initializations of both the random features and the BatchNorm parameters. We find that initializing $\beta$ to 1 improves the performance of training only BatchNorm.

**Appendix F.** Further experiments on training a random number of parameters per channel: (1) determining the number of random parameters necessary to reach the same performance as training $\gamma$ and $\beta$ and (2) training random parameters and the output layer.

**Appendix G.** Examining the role of making shortcut parameters trainable.

**Appendix H.** The distributions of $\gamma$ and $\beta$ for all networks as presented in Section 5 for ResNet-110 and ResNet-101.

**Appendix I.** Verifying that values of $\gamma$ that are close to zero can be set to 0 without affecting accuracy, meaning these features are not important to the learned representations.

**Appendix J.** The frequency with which activations are disabled for all ResNets that we study (corresponding to the activation experiments in Section 5).

## A  FORMAL RESTATEMENT OF BATCHNORM

The following is the batch normalization algorithm proposed by Ioffe & Szegedy (2015).

1. Let $x^{(1)}, ..., x^{(n)}$ be the pre-activations for a particular unit in a neural network for inputs 1 through $n$ in a mini-batch.

2. Let $\mu = \frac{1}{n} \sum_{i=1}^{n} x^{(i)}$

3. Let $\sigma^2 = \frac{1}{n} \sum_{i=1}^{n} (x^{(i)} - \mu)^2$

4. The batch-normalized pre-activation $\hat{x}^{(i)} = \gamma \frac{x^{(i)} - \mu}{\sqrt{\sigma^2}} + \beta$ where $\gamma$ and $\beta$ are trainable parameters.

5. The activations are $f(\hat{x}^{(i)})$ where $f$ is the activation function.

## B  DETAILS OF RESNETS

### B.1  CIFAR-10 RESNETS

**ResNet architectures.** We use the ResNets for CIFAR-10 as described by He et al. (2015a). Each network has an initial 3x3 convolutional layer from the three input channels to $16W$ channels (where $W$ is the width scaling factor). Afterwards, the network contains $3N$ residual blocks.

Each block has two 3x3 convolutional layers surrounded by a shortcut connection with the identity function and no trainable parameters. The first set of $N$ blocks have $16W$ filters, the second set of $N$ blocks have $32W$ filters, and the third set of $N$ blocks have $64W$ filters. The first layer in each set blocks downsamples by using a stride of 2; the corresponding shortcut connection has a 1x1 convolution that also downsamples. If there is a 1x1 convolution on the shortcut connection, it undergoes BatchNorm separately from the second convolutional layer of the block; the values are added together after normalization.

After the convolutions, each remaining channel undergoes average pooling and a fully-connected layer that produces ten output logits. Each convolutional layer is followed by batch normalization *before* the activation function is applied (He et al., 2016).

The depth is computed by counting the initial convolutional layer (1), the final output layer (1), and the layers in the residual blocks ($3N$ blocks $\times$ 2 layers per block). For example, when $N = 5$, there are 32 layers (the initial convolutional layer, the final output layer, and 30 layers in the blocks).

When the width scaling factor $W = 1$, we refer to the network as ResNet-*depth*, e.g., ResNet-32.

When the width scaling factor $W > 1$, we refer to the network as WRN-*depth-W*, e.g., WRN-14-4.

**Hyperparameters.** We initialize all networks using He normal initialization (He et al., 2015b), although we experiment with other initializations in Appendix E. The $\gamma$ parameters of BatchNorm are sampled uniformly from $[0, 1]$ and the $\beta$ parameters are set to 0. We train for 160 epochs with SGD with momentum (0.9) and a batch size of 128. The initial learning rate is 0.1 and drops by 10x at epochs 80 and 120. We perform data augmentation by normalizing per-pixel, randomly flipping horizontally, and randomly translating by up to four pixels in each direction. We use standard weight decay of 1e-4 on all parameters, including BatchNorm.

### B.2  IMAGENET RESNETS

**ResNet architectures.** We use the ResNets for ImageNet as described by He et al. (2015a). Each model has an initial 7x7 convolutional layer from three input channels to three output channels. Afterwards, there is a 3x3 max-pooling layer with a stride of 2.

Afterwards, the network has four groups of blocks with $N_1$, $N_2$, $N_3$, and $N_4$ blocks in each group. The first group of blocks has convolutions with 64 channels, the second group of blocks has convolutions with 128 channels, the third group of blocks has convolutions with 256 channels, and the fourth group of blocks has convolutions with 512 channels.

After the convolutions, each channel undergoes average pooling and a fully-connected layer that produces one thousand output logits. Each convolutional layer is followed by batch normalization *before* the activation function is applied (He et al., 2016).

The structure of the blocks differs; ResNet-18 and ResNet-34 have one block structure (a *basic block*) and ResNet-50, ResNet-101, and ResNet-200 have another block structure (a *bottleneck* block). These different block structures mean that ResNet-18 and ResNet-34 have a different number of output and shortcut parameters than the other ResNets. The basic block is identical to the block in the CIFAR-10 ResNets: two 3x3 convolutional layers (each followed by BatchNorm and a ReLU activation). The bottleneck block comprises a 1x1 convolution, a 3x3 convolution, and a final 1x1 convolution that increases the number of channels by 4x; the first 1x1 convolution in the next block decreases the number of channels by 4x back to the original value. In both cases, if the block downsamples the number of filters, it does so by using stride 2 on the first 3x3 convolutional layer and adding a 1x1 convolutional layer to the skip connection.

The depth is computed just as with the CIFAR-10 ResNets: by counting the initial convolutional layer (1), the final output layer (1), and the layers in the residual block. We refer to the network as ResNet-*depth*, e.g., ResNet-50.

The table below specifies the values of $N_1$, $N_2$, $N_3$, and $N_4$ for each of the ResNets we use. These are the same values as specified by He et al. (2015a).

| Name | $N_1$ | $N_2$ | $N_3$ | $N_4$ |
|---|---|---|---|---|
| ResNet-18 | 2 | 2 | 2 | 2 |
| ResNet-34 | 3 | 4 | 6 | 3 |
| ResNet-50 | 3 | 4 | 6 | 3 |
| ResNet-101 | 3 | 4 | 23 | 3 |
| ResNet-200 | 3 | 24 | 36 | 3 |

**Hyperparameters.** We initialize all networks using He normal initialization (He et al., 2015b). The $\gamma$ parameters of BatchNorm are sampled uniformly from $[0, 1]$ and the $\beta$ parameters are set to 0. We train for 90 epochs with SGD with momentum (0.9) and a batch size of 1024. The initial learning rate is 0.4 and drops by 10x at epochs 30, 60, and 80. The learning rate linearly warms up from 0 to 0.4 over the first 5 epochs. We perform data augmentation by normalizing per-pixel, randomly flipping horizontally, and randomly selecting a crop of the image with a scale between 0.1 and 1.0 and an aspect ratio of between 0.8 and 1.25. After this augmentation, the image is resized to 224x224. We use standard weight decay of 1e-4 on all parameters, including BatchNorm.

## C  RESULTS FOR VGG ARCHITECTURE

In this Section, we repeat the major experiments from the main body of the paper for VGG-style neural networks (Simonyan & Zisserman, 2014) for CIFAR-10. The particular networks we use were adapted for CIFAR-10 by Liu et al. (2019). The distinguishing quality of these networks is that they do not have residual connections, meaning they provide a different style of architecture in which to explore the role of the BatchNorm parameters and the performance of training only BatchNorm.

### C.1  ARCHITECTURE AND HYPERPARAMETERS

**Architecture.** We consider four VGG networks: VGG-11, VGG-13, VGG-16, and VGG-19. Each of these networks consists of a succession of 3x3 convolutional layers (each followed by BatchNorm) and max-pooling layers with stride 2 that downsample the activation maps. After some max-pooling layers, the number of channels per layer sometimes doubles. After the final layer, the channels are combined using average pooling and a linear output layer produces ten logits.

The specific configuration of each network is below. The numbers are the number of channels per layer, and $M$ represents a max-pooling layer with stride 2.

| Name | Configuration |
|---|---|
| VGG-11 | 64, $M$, 128, $M$, 256, 256, $M$, 512, 512, $M$, 512, 512 |
| VGG-13 | 64, 64, $M$, 128, 128, $M$, 256, 256, $M$, 512, 512, $M$, 512, 512 |
| VGG-16 | 64, 64, $M$, 128, 128, $M$, 256, 256, 256, $M$, 512, 512, 512, $M$, 512, 512, 512 |
| VGG-19 | 64, 64, $M$, 128, 128, $M$, 256, 256, 256, 256, $M$, 512, 512, 512, 512, $M$, 512, 512, 512, 512 |

**Hyperparameters.** We initialize all networks using He normal initialization. The $\gamma$ parameters of BatchNorm are sampled uniformly from $[0, 1]$ and the $\beta$ parameters are set to 0. We train for 160 epochs with SGD with momentum (0.9) and a batch size of 128. The initial learning rate is 0.1 and drops by 10x at epochs 80 and 120. We perform data augmentation by normalizing per-pixel, randomly flipping horizontally, and randomly translating by up to four pixels in each direction. We use standard weight decay of 1e-4 on all parameters, including BatchNorm.

**Parameter-counts.** Below are the number of parameters in the entire network, the BatchNorm layers, and the output layer. This table corresponds to Table 1.

| Family | VGG for CIFAR-10 | | | |
|---|---|---|---|---|
| Depth | 11 | 13 | 16 | 19 |
| Width Scale | 1 | 1 | 1 | 1 |
| Total | 9.23M | 9.42M | 14.73M | 20.04M |
| BatchNorm | 5.5K | 5.89K | 8.45K | 11.01K |
| Output | 5.13K | 5.13K | 5.13K | 5.13K |
| BatchNorm | 0.06% | 0.06% | 0.06% | 0.05% |
| Output | 0.06% | 0.05% | 0.03% | 0.03% |

## C.2 RESULTS

In this subsection, we compare the behavior of the VGG family of networks to the major results in the main body of the paper. Unlike the ResNets, disabling $\gamma$ and $\beta$ has no effect on the performance of the VGG networks when all other parameters are trainable (Figure 8, corresponding to Figure 1 in the main body).

When training only BatchNorm (Figure 9, corresponding to Figure 2 in the main body), the VGG networks reach between 57% (VGG-11) and 61% (VGG-19) accuracy, lower than full performance (92% to 93%) but higher than chance (10%). Since these architectures do not have skip connections, we are not able to explore as wide a range of depths as we do with the ResNets. However we do see that increasing the number of available features by increasing depth results in higher accuracy when training only BatchNorm. Making the output layer trainable improves performance by more than 3 percentage points for VGG-11 but less than 2 percentage points for VGG-16 and VGG-19, the same limited improvements as in the CIFAR-10 ResNets. Training the output layer alone performs far worse: 41% accuracy for VGG-11 dropping down to 27% for VGG-19.

The performance of training only BatchNorm remains far higher than training two random parameters per channel. Accuracy is higher by between 13 and 14 percentage points, again suggesting that $\gamma$ and $\beta$ have particular expressive power as a per-channel coefficient and bias.

As we observed in Section 5 for the ResNets, training only BatchNorm results in a large number of $\gamma$ parameters close to zero (Figure 21). Nearly half (44% for VGG-11 and 48% for VGG-19) of all $\gamma$ parameters have magnitudes less than 0.01 when training only BatchNorm as compared to between 2% and 3% when all parameters are trainable.

## D COMPARING TRAINING ONLY BATCHNORM TO SMALL RESNETS

As another baseline to contextualize our findings, we compare training only BatchNorm to training all parameters in small ResNets (Figure 10). This experiment assesses whether, for a fixed budget of trainable parameters, it is better to train a large number of random features or a small number of learnable features. To make this comparison, we perform grid search over depths and widths to find ResNets whose total parameter-counts are similar to the BatchNorm parameter-counts in

our networks. The small ResNets indeed outperform training only BatchNorm, at best by 5 to 10 percentage points.[9] In other words, for a fixed trainable parameter budget, learning features indeed outperforms shifting and scaling random features with BatchNorm. (We emphasize that we intend this experiment only to contextualize our earlier findings; our goal is to study the role and expressive power of $\gamma$ and $\beta$, not to find the most performant way to allocate a fixed parameter budget.)

# E   VARYING INITIALIZATION

## E.1   FEATURE INITIALIZATION

In the main body of the paper, we show that ResNets comprising shifts and rescalings of random features can reach high accuracy. However, we have only studied one class of random features: those produced by He normal initialization (He et al., 2015b). It is possible that other initializations may produce features that result in higher accuracy. We explored initializing with a uniform distribution, binarized weights, and samples from a normal distribution that were orthogonalized using SVD, however, doing so had no discernible effect on accuracy.

## E.2   BATCHNORM INITIALIZATION

Similarly, our BatchNorm initialization scheme ($\gamma \sim \mathcal{U}[0, 1]$, $\beta = 0$) was designed with training all parameters in mind. It is possible that a different scheme may be more suitable when training only BatchNorm. We studied three alternatives: another standard practice for BatchNorm ($\gamma = 1$, $\beta = 0$), centering $\gamma$ around 0 ($\gamma \sim \mathcal{U}[-1, 1]$, $\beta = 0$), and ensuring a greater fraction of normalized features pass through the ReLU activation ($\gamma = 1$, $\beta = 1$). The first two schemes did not change performance.

The third, where $\beta = 1$, increased the accuracy of the BatchNorm-only experiments by 1 to 3 percentage points across all depths and widths (Figure 11), improving ResNet-866 to 84% accuracy, WRN-14-32 to 75%, and ResNet-200 to 37% top-5 accuracy. Interestingly, doing so *lowered* the accuracy when all parameters are trainable by 3.5% on WRN-14-32 and 0.5% (top-5) on ResNet-200; on the deepest networks for CIFAR-10, it caused many runs to fail entirely.

We conclude that (1) ideal initialization schemes for the BatchNorm parameters in the BatchNorm-only and standard scenarios appear to be different and (2) the standard training regime is indeed sensitive to the choice of BatchNorm initializations.

# F   FREEZING PARAMETERS

In Section 4, we tried training two random parameters per channel in place of $\gamma$ and $\beta$ to study the extent to which the BatchNorm-only performance was due merely to the aggregate number of trainable parameters. We found that training two random parameters per channel resulted in far lower accuracy, suggesting that $\gamma$ and $\beta$ indeed have particular expressive power as a product of their position as a per-feature coefficient and bias.

In Figure 13, we study the number of number of random parameters that must be made trainable per-channel in order to match the performance of training only BatchNorm. (Note: we only collected this data for the CIFAR-10 Networks.) For the shallower ResNets, we must train 8 random parameters per channel (4x as many parameters as the number of BatchNorm parameters) to match the performance of training only $\gamma$ and $\beta$. For the deeper ResNets, we must train 16 random parameters per channel (8x as many parameters as the number of BatchNorm parameters) to match the performance of training only $\gamma$ and $\beta$. These results further support the belief that $\gamma$ and $\beta$ have greater expressive power than other parameters in the network.

In Figure 14, we study training both two random parameters per channel and the output layer. On CIFAR-10, doing so matches or underperforms training only BatchNorm. On ImageNet, doing so outperforms training only BatchNorm in shallower networks and matches the performance of training only BatchNorm in deeper networks. In all cases, it far underperforms training BatchNorm and the output layer.

---

[9]These networks separate into two accuracy strata based on width. The lower stratum has width scale 1/8 and is deeper, while the higher stratum has width scale 1/4 to 1/2 and is shallower.

## G MAKING SHORTCUTS TRAINABLE

It is possible to train deep ResNets due to shortcut connections that propagate gradients to the earlier layers (He et al., 2015a; Balduzzi et al., 2017). Nearly all shortcuts use the identity function and have no trainable parameters. However, the shortcuts that downsample use 1x1 convolutions. It is possible that, by freezing these parameters, we have inhibited the ability of our networks to propagate gradients to lower layers and take full advantage of the BatchNorm parameters.

To evaluate the impact of this restriction, we enable training for the shortcut and output layer parameters in addition to the BatchNorm parameters (Figure 12). For ResNet-110, making these additional 3.2k (0.38%) parameters trainable improves accuracy by five points to 74.6%, suggesting that freezing these parameters indeed affected performance. However, on deeper networks, the returns from making shortcut and output parameters trainable diminish, with no improvement in accuracy on ResNet-866. If freezing these parameters were really an impediment to gradient propagation, we would expect the deepest networks to benefit most, so this evidence does not support our hypothesis.

As an alternate explanation, we propose that accuracy improves simply due to the presence of more trainable parameters. As evidence for this claim, the shallowest networks—for which shortcut and output parameters make up a larger proportion of weights—benefit most when these parameters are trainable. Doing so quadruples the parameter-count of ResNet-14, which improves from 48% to 63% accuracy, and adds a further 2.6M parameters (315x the number of BatchNorm parameters) to WRN-14-32, which improves from 73% to 87% accuracy. We see a similar effect for the ImageNet networks: the top-5 accuracy of ResNet-101 improves from 25% to 72%, but the number of parameters also increases 26x from 105K to 2.8M. Finally, if we freeze BatchNorm and train only the shortcuts and outputs, performance of shallower networks is even better than training just BatchNorm, reaching 49% (vs. 48%) for ResNet-14 and 35% (vs. 25%) top-5 for ResNet-101.

## H BATCHNORM PARAMETER DISTRIBUTIONS FOR ALL NETWORKS

In Section 5, we plot the distributions of $\gamma$ and $\beta$ for ResNet-110 for CIFAR-10 and ResNet-101 for ImageNet. For all other networks, we include a summary plot (Figure 5) showing the fraction of $\gamma$ parameters below various thresholds. In this Appendix, we plot the distributions for the deep ResNets for CIFAR-10 (Figures 15 and 16), the wide ResNets for CIFAR-10 (Figures 17 and 18), the ImageNet ResNets (Figures 19 and 20), and the VGG networks (Figures 21 and 22).

## I FREQUENCY OF ZERO ACTIVATIONS

In Section 5, we plot the number of ReLUs for which the $\Pr[\text{activation} = 0] > 0.99$ for each ResNet. In Figure 23 (deep ResNets for CIFAR-10), Figure 24 (wide ResNets for CIFAR-10), and Figure 25 (ImageNet ResNets), we plot a histogram of the values of $\Pr[\text{activation} = 0]$ for the ReLUs in each network. In general, training only BatchNorm leads to many activations that are either always off or always on. In contrast, training all parameters leads to many activations that are on some of the time (10% to 70%).

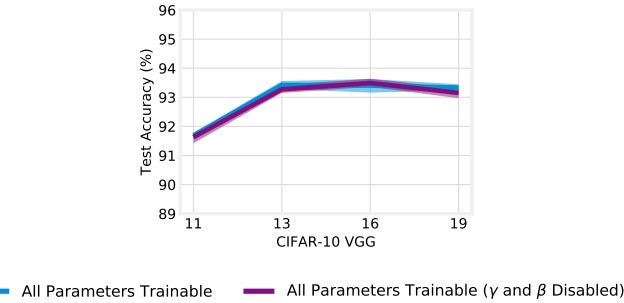

Figure 8: Accuracy when training VGG networks. Accuracy does not differ when training with $\gamma$ and $\beta$ disabled.

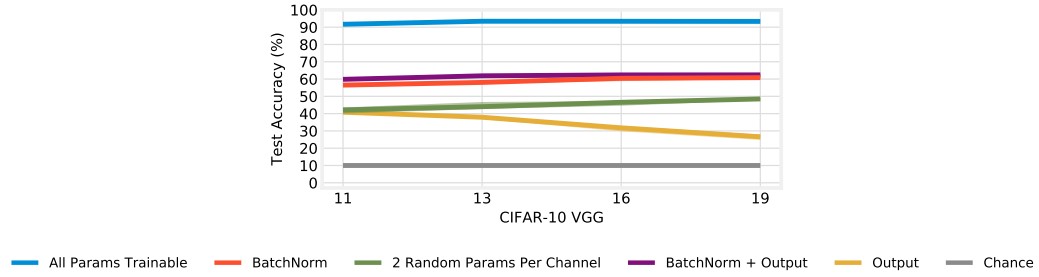

Figure 9: Accuracy of VGG networks for CIFAR-10 when making certain parameters trainable.

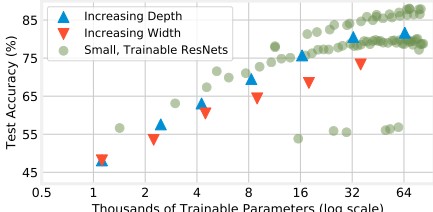

Figure 10: Comparing training only BatchNorm with training all parameters in small ResNets found via grid search.

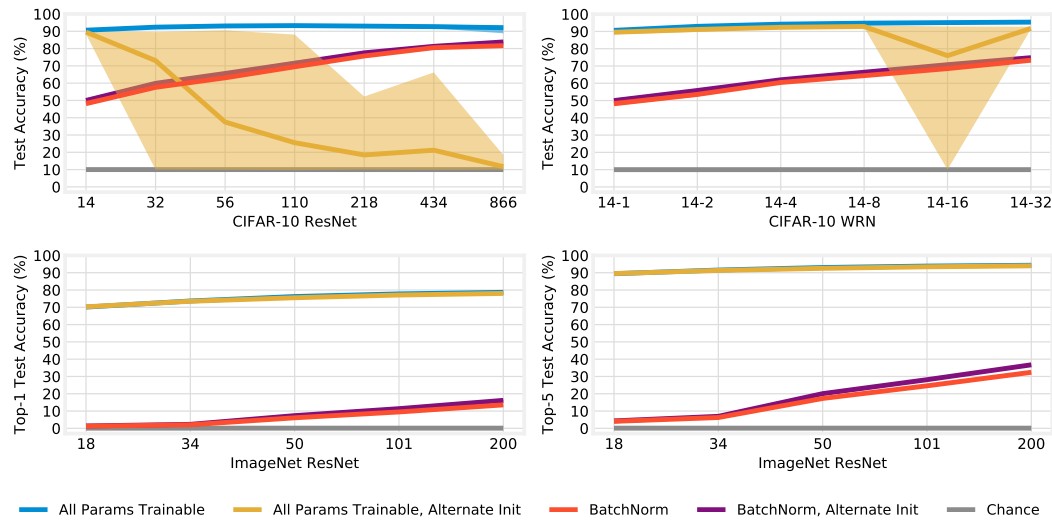

Figure 11: Accuracy of ResNets for CIFAR-10 (top left, deep; top right, wide) and ImageNet (bottom left, top-1 accuracy; bottom right, top-5 accuracy) with the original BatchNorm initialization ($\gamma \sim \mathcal{U}[0,1]$, $\beta = 0$) and an alternate initialization ($\gamma = 1$, $\beta = 1$).

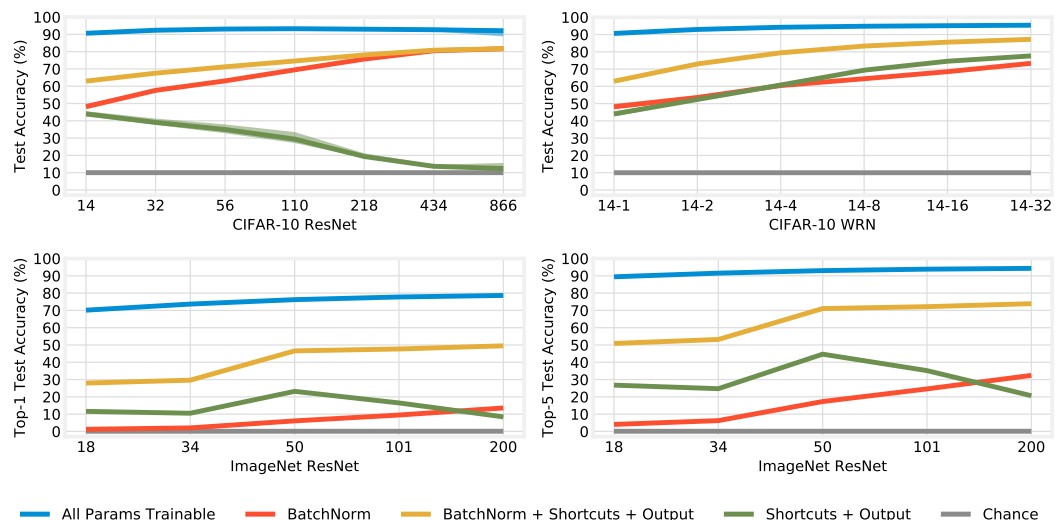

Figure 12: Accuracy of ResNets for CIFAR-10 (top left, deep; top right, wide) and ImageNet (bottom left, top-1 accuracy; bottom right, top-5 accuracy) when making output and shortcut layers trainable in addition to BatchNorm.

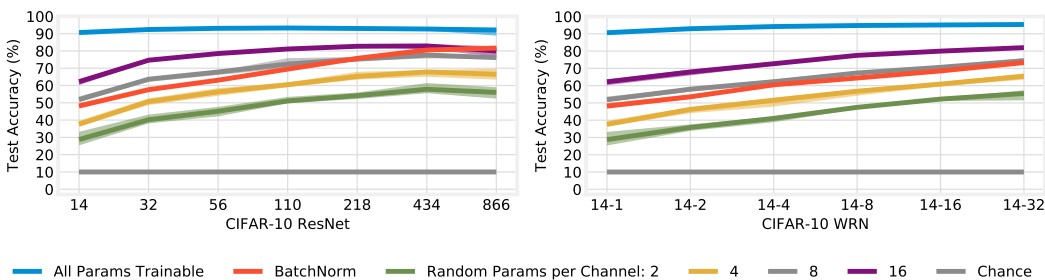

Figure 13: Accuracy of ResNets for CIFAR-10 (left, deep; right, wide) when training only a certain number of randomly-selected parameters per convolutional channel. When training two random parameters per-channel, we are training the same number of parameters as when training only BatchNorm.

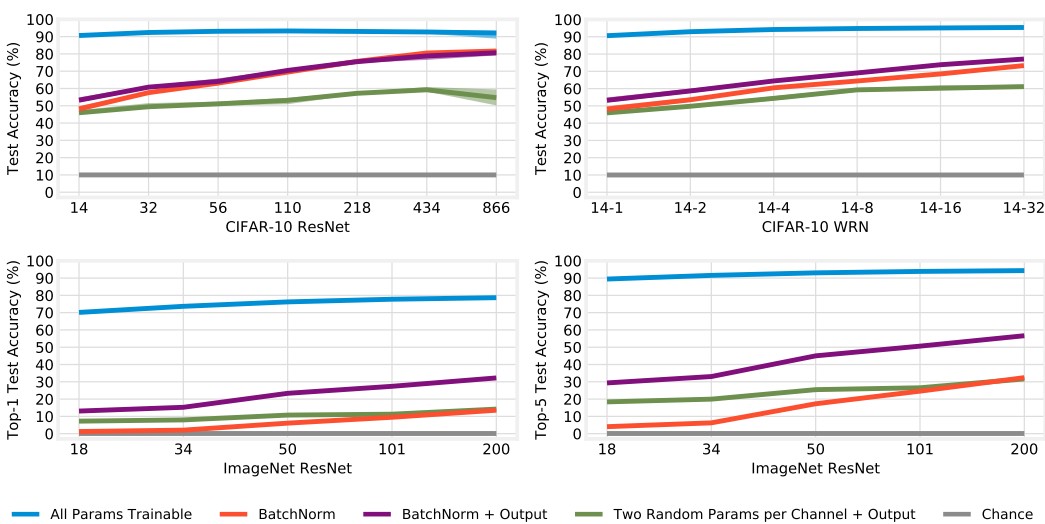

Figure 14: Accuracy of ResNets for CIFAR-10 (top left, deep; top right, wide) and ImageNet (bottom left, top-1 accuracy; bottom right, top-5 accuracy) when making two random parameters per channel and the output layer trainable.

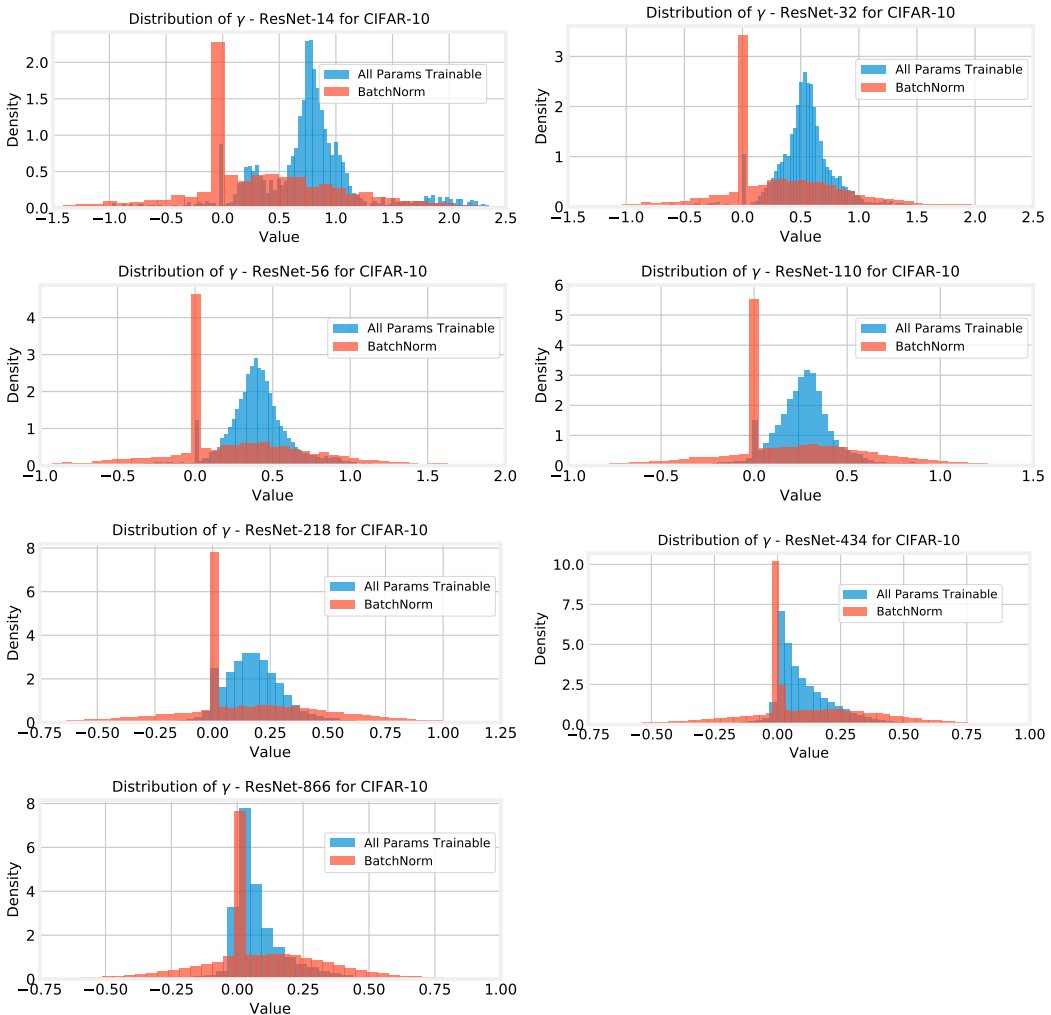

Figure 15: The distributions of $\gamma$ for the deep CIFAR-10 ResNets.

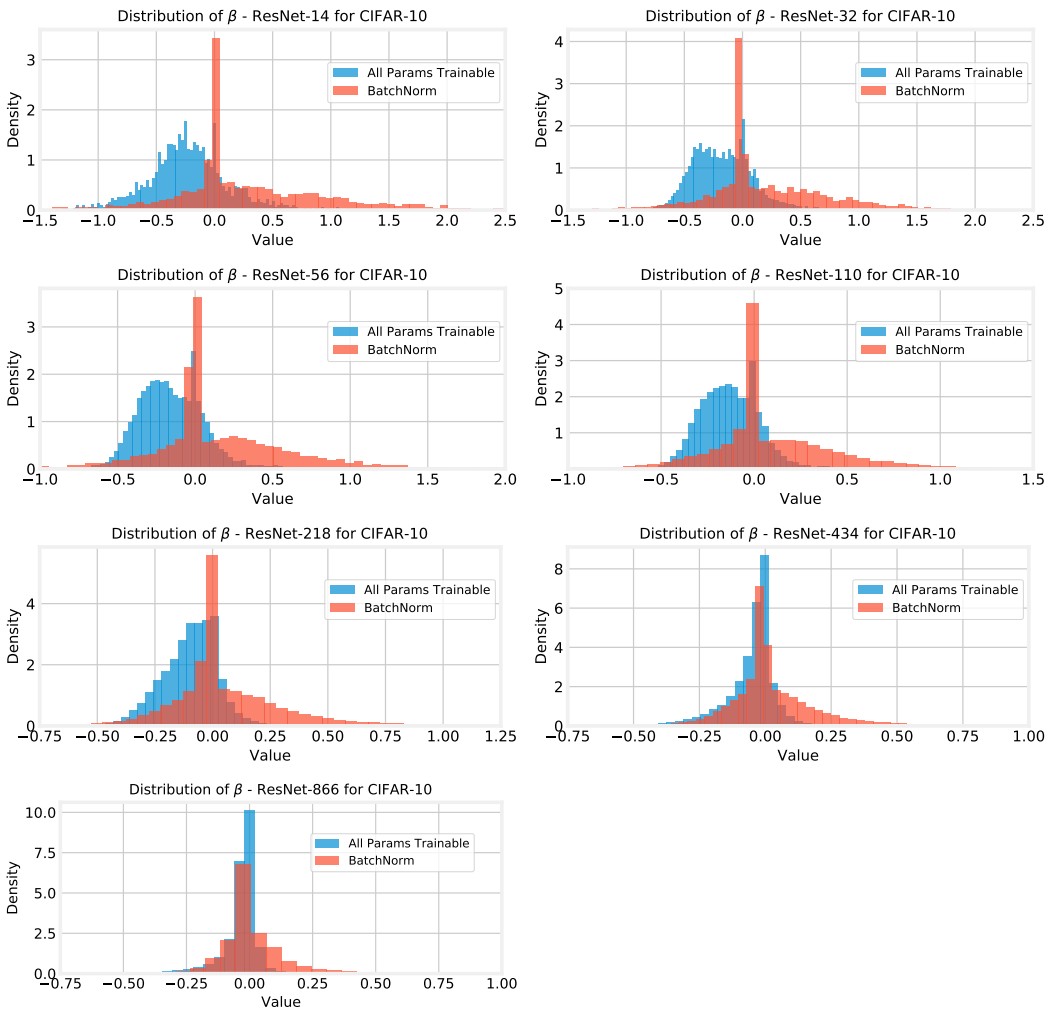

Figure 16: The distributions of $\beta$ for the deep CIFAR-10 ResNets.

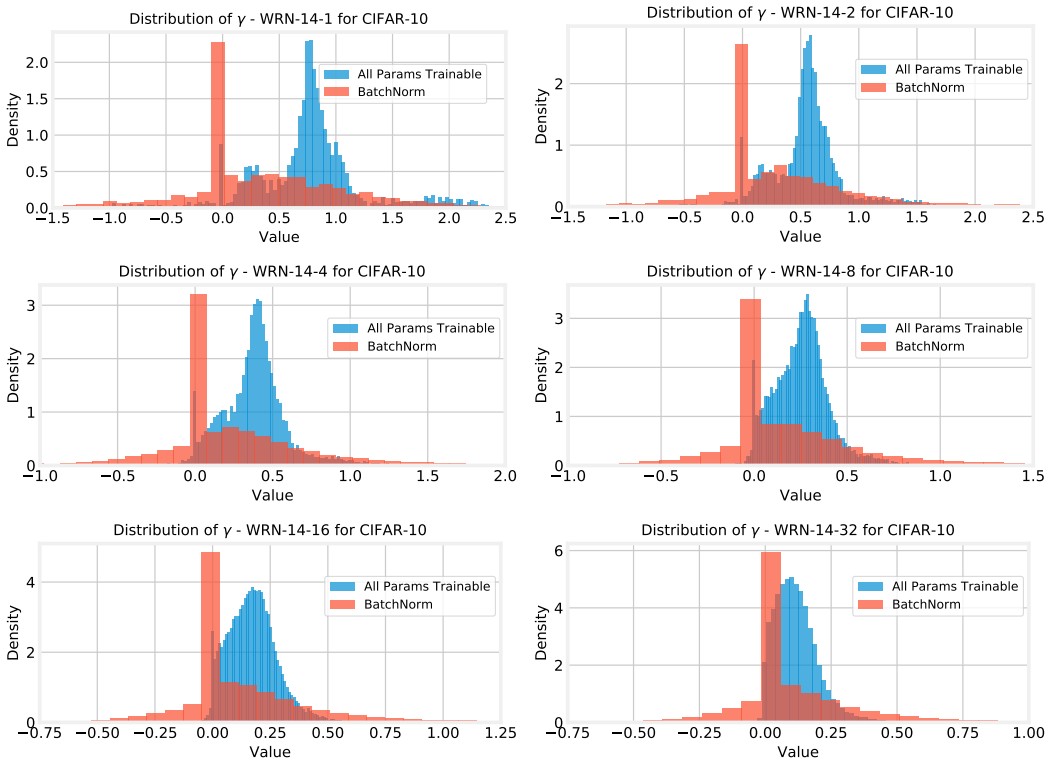

Figure 17: The distributions of $\gamma$ for the wide CIFAR-10 ResNets.

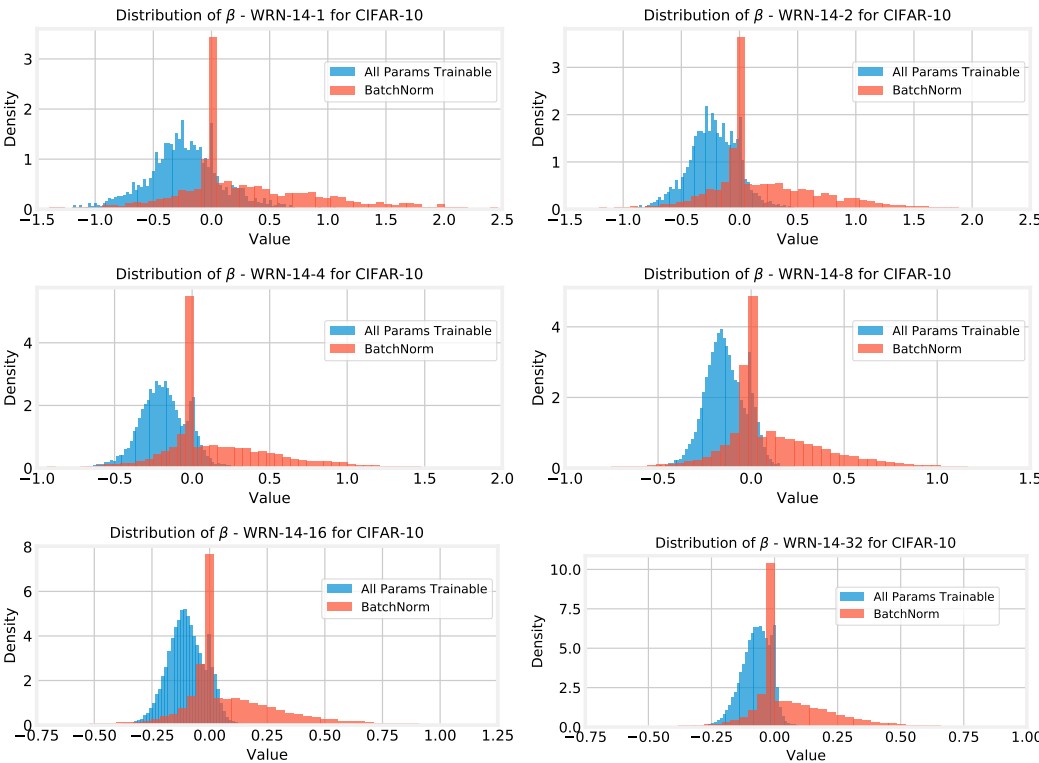

Figure 18: The distributions of $\beta$ for the wide CIFAR-10 ResNets.

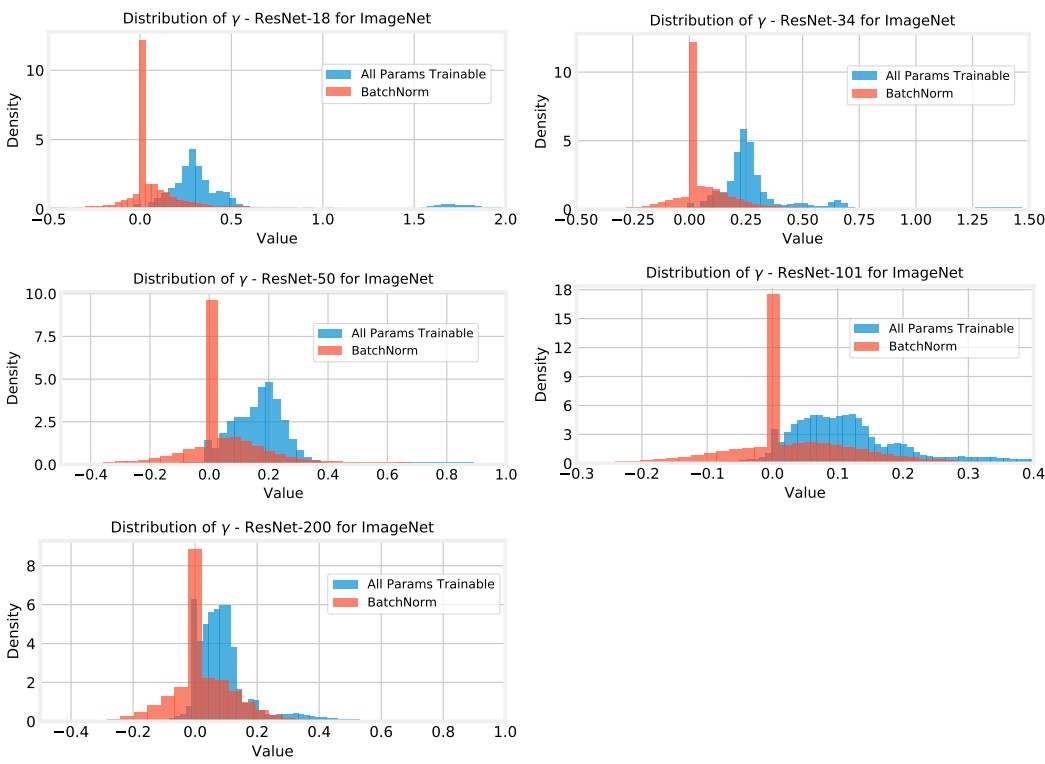

Figure 19: The distributions of $\gamma$ for the ImageNet ResNets.

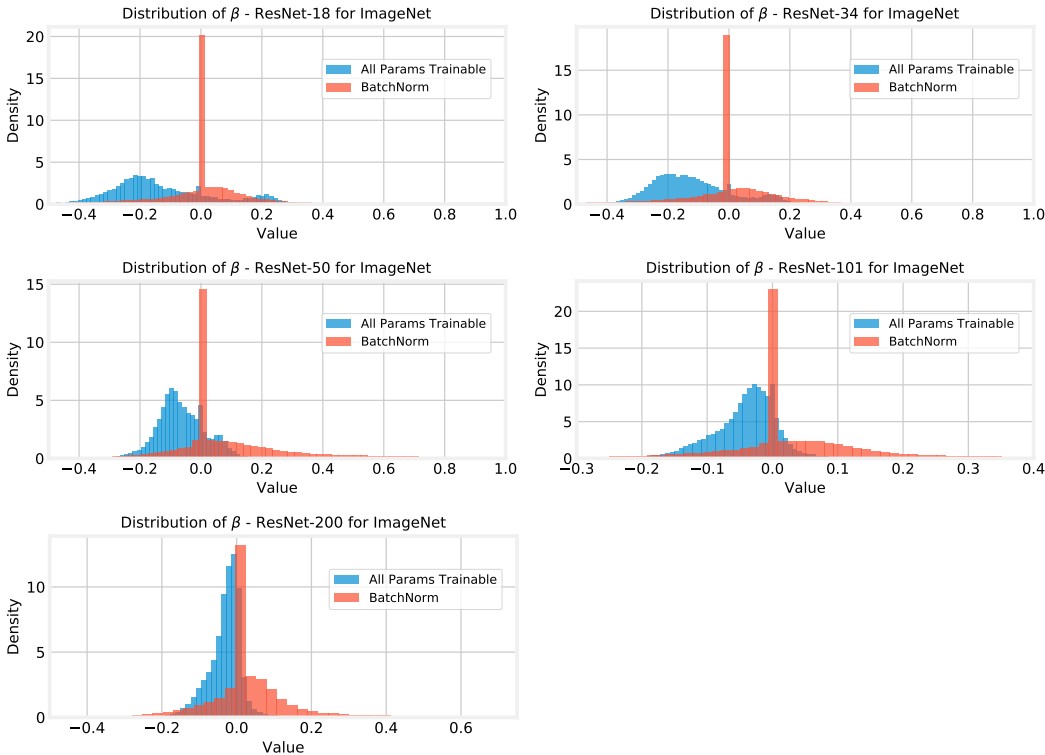

Figure 20: The distributions of $\beta$ for the ImageNet ResNets.

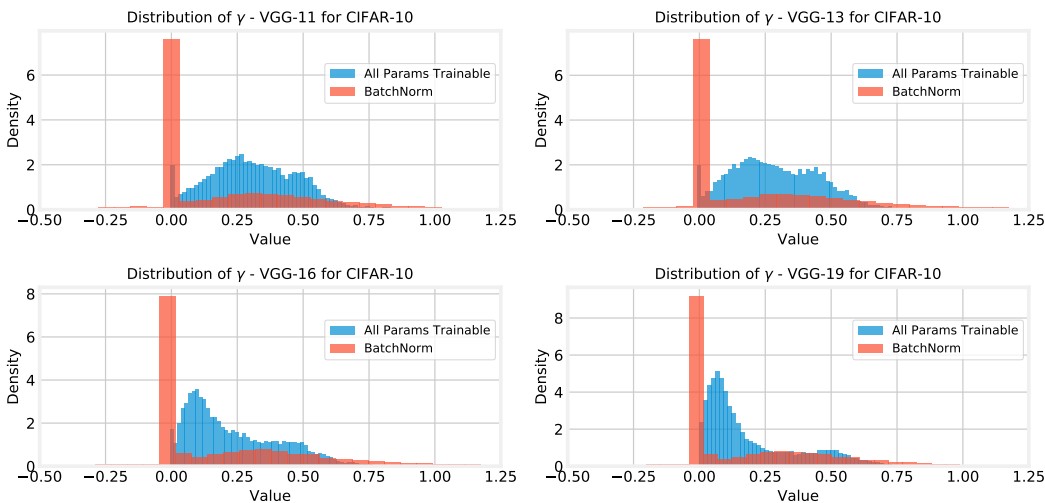

Figure 21: The distributions of $\gamma$ for the VGG networks.

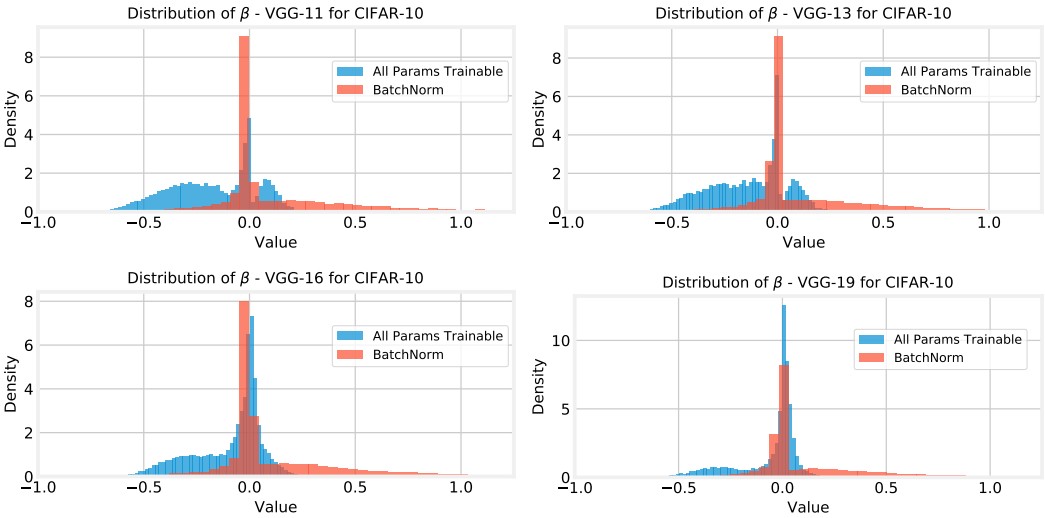

Figure 22: The distributions of $\beta$ for the VGG networks.

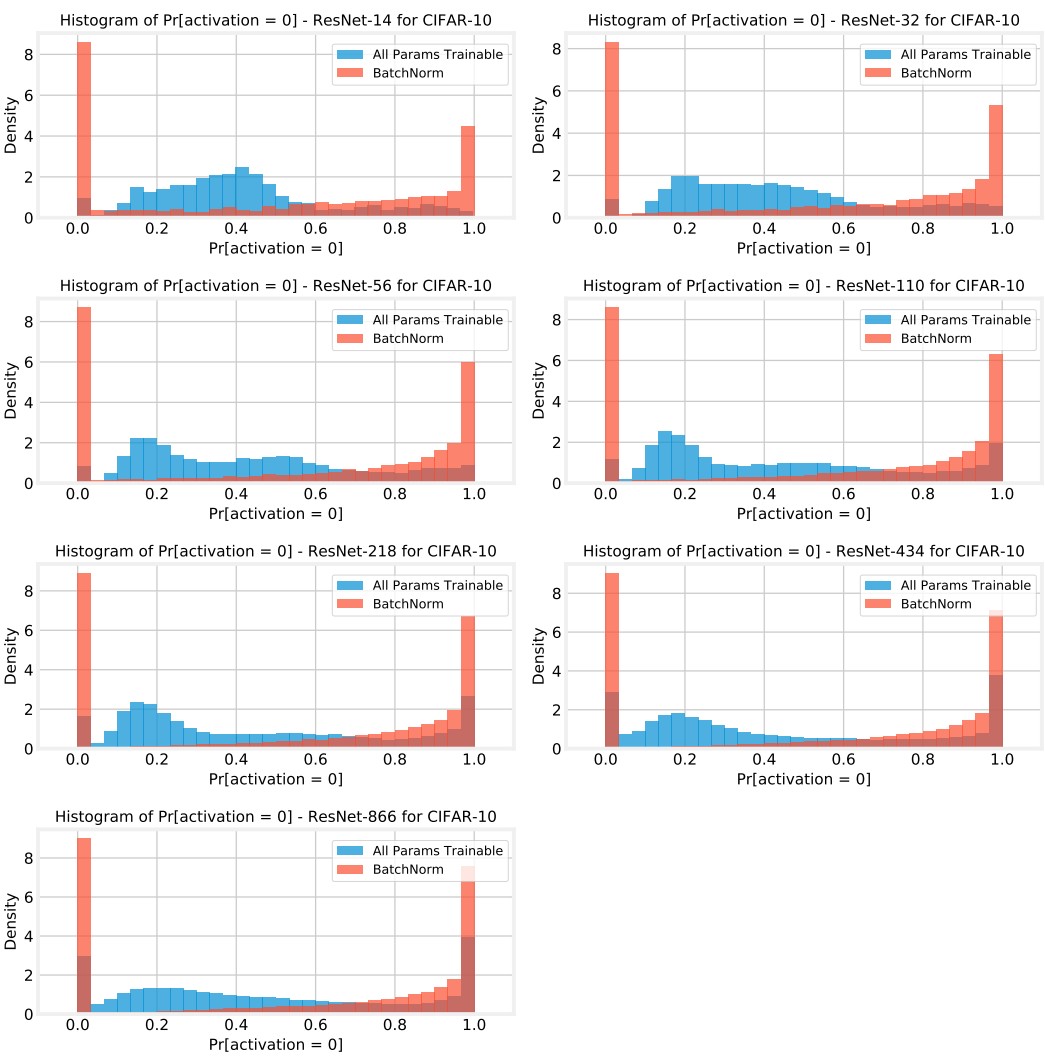

Figure 23: Per-ReLU activation freqencies for deep CIFAR-10 ResNets.

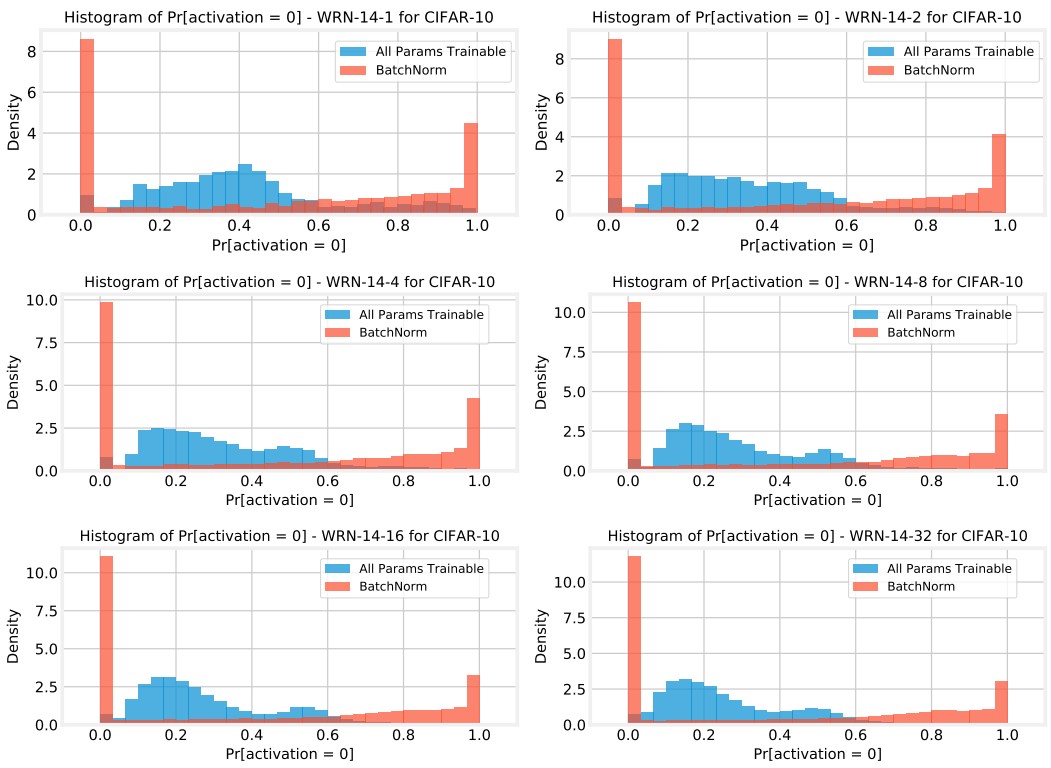

Figure 24: Per-ReLU activation freqencies for CIFAR-10 WRNs.

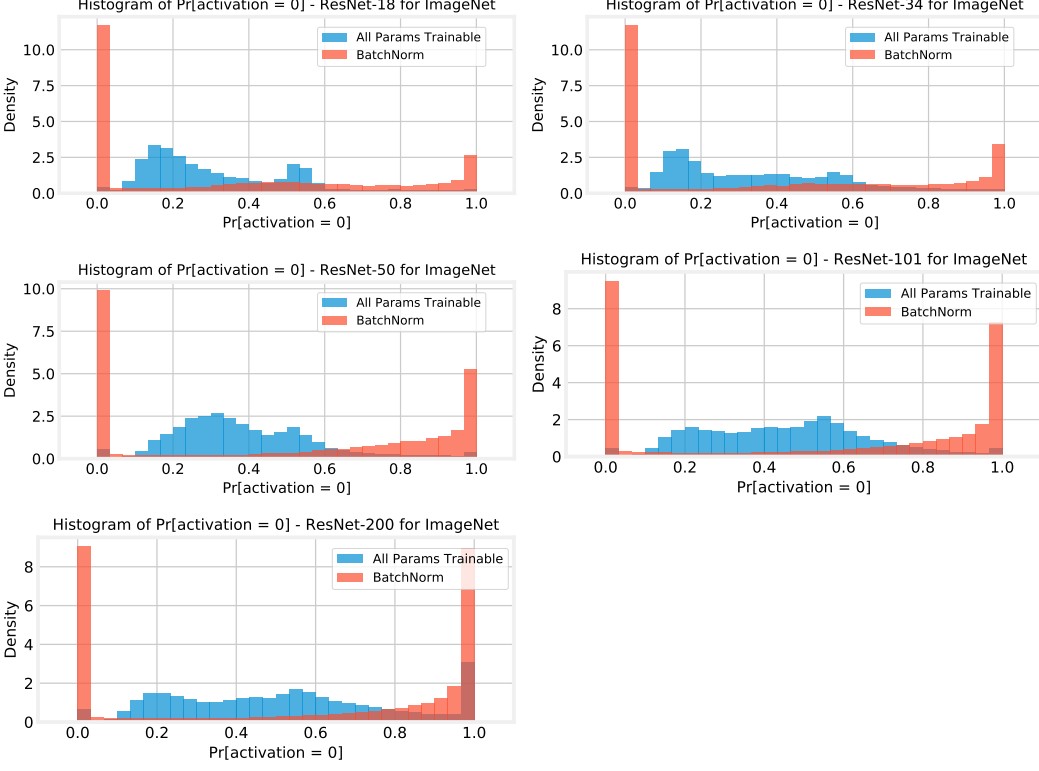

Figure 25: Per-ReLU activation freqencies for ImageNet ResNets.

