# OpenReview forum: "Training BatchNorm and Only BatchNorm: On the Expressive Power of Random Features in CNNs"
_ICLR.cc/2021/Conference — ICLR 2021 Poster_

### Official Review · AnonReviewer4 · 2020-10-21
**This paper studies the expressive power of batchnorm parameters by training only these parameters while fixing other randomly initialized parameters.**

**Rating:** 6
**Confidence:** 3

**Review:**

This paper studies the expressive power of batchnorm parameters by training only these parameters while fixing other randomly initialized parameters. With experiments on different datasets and models, the authors show that batchnorm parameters are consistently more expressive than other parameters. The authors also try to explain such phenomenon by examining the values of parameters and activations, showing that training BN only can lead to sparse values.

Overall, I am leaning toward accepting. The following are the pros and cons of this paper.

Strengths:
1. The paper is well written, and it is easy to follow.
2. Understanding the mechanism behind batchnorm(BN) is a very important problem.
3. The paper provides comprehensive experiments, demonstrating the expressive power of BN parameters from small models to large models (controlled by both depth and width).
4. To explain the effect, the paper provides some interesting results from training BN parameters only: there is sparsity in the BN parameters, and sparsity in ReLU activations.

Weaknesses:
1. My major concern is that the expressive power of BN parameters has already been discussed in several papers [1,2] on multi-task learning or transfer learning. For example, [1] showed training different BN parameters for different tasks can greatly enhance the models' performance for multiple tasks without significantly increasing the model size. I think this paper should take the literature into the discussion and better pose the contribution of this paper.
2. I think the paper could be stronger if it could provide more implication of its findings. More discussions on how can we make use of these findings, (one example would be like in [1] we can build a compact model for multiple tasks) would greatly enhance the impact.
3. This paper focuses on batchnorm. Actually, layernorm (in transformer), instance norm, group norm and so on, all have such parameters. It would be interesting to see whether the results generalize to other normalization types.
4. It would also be interesting to discuss more on the difficulty of training BN parameters only compared with the difficulty of training whole networks. Does it become more sensitive or less sensitive to learning rate, batch size?


Comments:

1. Introduction, second paragraph, 'in the time since batchnorm was first proposed' -> 'since batchnorm was first proposed'.

[1] Mudrakarta, Pramod Kaushik and Sandler, Mark and Zhmoginov, Andrey and Howard, Andrew. K for the Price of 1: Parameter-efficient Multi-task and Transfer Learning, ICLR 2019.
[2] S.-A. Rebuffi, H. Bilen, and A. Vedaldi. Learning multiple visual domains with residual adapters. In NIPS, 2017.


***********************
After rebuttal: I maintain my rating and think this paper is on the borderline. I think the paper is interesting, but the novelty and significance is limited by previous works along the direction.

---

> ### Author Response · Authors · 2020-11-25
> **Author Response (Part 2)**
>
> _This paper focuses on BatchNorm. Actually, layernorm (in transformer), instance norm, group norm and so on, all have such parameters. It would be interesting to see whether the results generalize to other normalization types._
>
> We strongly agree that this is a valuable direction for future work (and we mention it as such at the end of the paper). This direction could elucidate whether our findings are specific to BatchNorm or whether they more generally apply to affine parameters applied to features. In this vein, we are also interested in running our experiment in contexts that do not have normalization (i.e., adding gamma and beta parameters to standard convnets or to ResNets with BatchNorm-free FixUp initialization [3]). Due to the substantial scale of the current paper, it was beyond our means and beyond our scope to conduct these experiments during the discussion period, but we believe this is an exciting direction for future work.
>
> ---
>
> _Does it become more sensitive or less sensitive to learning rate, batch size?_
>
> We agree that this is also a great question. It is possible that (as you suggest) these networks are less sensitive to hyperparameters. Moreover, it is possible that there may even be better hyperparameters that allow these BatchNorm-only networks to reach higher accuracy. Due to the substantial scope of the experiments in the current paper, we restricted our experiments to the standard hyperparameters for these networks. However, we think this is an exciting direction for future work and that there may be fruitful opportunities to improve upon our current findings.
>
> ---
> _Introduction, second paragraph._
>
> Thank you for the suggestion. We have made this change in our updated manuscript.
>
> [1] Pramod Kaushik Mudrarkarta et al. K for the Price of 1: Parameter-Efficient Multi-Task and Transfer Learning. ICLR 2019.
>
> [2] S.-A. Rebuffi et al. Learning multiple visual domains with residual adapters. NeurIPS 2017
>
> [3] Hongyi Zhang et al. FixUp initialization: Residual Learning Without Normalization. ICLR 2019.

---

> ### Author Response · Authors · 2020-11-25
> **Author Response (Part 1)**
>
> We thank the reviewer for the detailed feedback. We have responded to the reviewer’s comments below.
>
> ---
>
> _My major concern is that the expressive power of the BN parameters has already been discussed in several papers [1, 2] on multi-task learning or transfer learning. For example, [1] showed training different BN parameters for different tasks can greatly enhance the models’ performance for multiple tasks without significantly increasing model size. I think this paper should take the literature into the discussion and better pose the contribution of this paper._
>
> Based on your feedback and suggestions for related work, we have made substantial revisions to the abstract, introduction, and discussion to better scope our contribution. In particular, we have framed our contribution as a way to study the expressive power of per-feature affine parameters; freezing all other weights at their random initializations makes it possible to distinguish the contribution of these affine parameters from the trained features that they typically transform, and BatchNorm is a convenient vantage point from which to study these parameters. In this way, our work complements the related work you suggest while providing a novel perspective on these affine parameters.
>
> We have also updated our related work section to discussing the relationship between our research and the related work you mention. In particular, we have added a subsection on “exploiting the expressive power of affine transformations” to discuss other instances in the literature where per-feature affine transformations are a key element of various techniques (including [1] and [2]). We have also updated the related work section on “training only batchnorm” to mention [1], since it briefly runs the same experiment we do on a MobileNet.
>
> ---
>
> _I think the paper could be stronger if it could provide more implication of its findings. More discussions on how can we make use of these findings (one example would be like in [1] we can build a compact model for multiple tasks) would greatly enhance the impact._
>
> We really appreciate your suggestion of looking at [1] in this context. Indeed, we believe our work shows that there is an opportunity to produce compact, multi-task networks. The benefit of our approach over that applied in [1] is that the non-BatchNorm weights in the network are random. This means that we do not need to remember the state of the network; only the random seed used to produce it. As such, we envision the following application:
>
> 1. To train the network, use a random seed to generate the weights and train separate BatchNorm parameters (and optionally an output layer) for each task (in the same manner as [1]).
> 2. Use the network for inference, use the random seed to reconstruct the weights and load the proper BatchNorm parameters (and optionally the output layer).
>
> The advantage of this approach is that there is no need to store the weights of the network. As our experiments show, training only BatchNorm benefits significantly from larger-scale networks. In the approach proposed in [1], these networks would be expensive to store; however, since we only need to store the random seed, storage size is constant regardless of the size of the network. As an additional opportunity for future work, it is possible that different initialization distributions may be well-suited for different tasks; if this is the case, we can use a different (distribution, seed) pair for each task.
>
> We have added the description of this application to our discussion section (Section 6 Paragraph 4).

---

### Official Review · AnonReviewer1 · 2020-10-24
**Review of "Training BatchNorm and Only BatchNorm"**

**Rating:** 6
**Confidence:** 5

**Review:**

Summary
-------------
The authors explore the representational power of BatchNorm's affine parameters (scale $\gamma$ and bias $\beta$). For that, they freeze the randomly-initialized parameters of different versions of ResNet and VGG, and only train the affine transformations. They also compare the expressiveness of BatchNorm coefficients with respect to the same amount of neural net parameters.  The main conclusions of this work are that BatchNorm coefficients have a greater discriminative power than the rest of network parameters. Moreover, in random networks, $\gamma$ seems to disable non-useful features, disabling more than 25% of the channels, and in non-random networks it may prevent overshooting. They also show how these coefficients interact with networks of different depth and width, concluding that deeper random networks achieve better performance than wider random networks with the same amount of BatchNorm parameters.

Overall Review
--------------------
Understanding the principles that make BatchNorm so successful is key to design better neural network architectures. This makes this submission interesting for the research community. On the other hand, it is known that $\gamma$ and $\beta$ are highly expressive [1,2] and that they produce sparsity [3]. In summary, in its current state, it is difficult to see what the impact of this submission in future works will be and I cannot recommend it for acceptance. Thus, I encourage the authors to increase the depth of their study (see weaknesses).

Strengths
-------------
* Understanding BatchNorm is an important and interesting topic.
* The paper is well-organized and easy to read.
* The proposed experiments are sound.

Weaknesses
-----------------
* It is known that BatchNorm coefficients have a high expressive power. For instance, FILM [1] layers modulate neural networks by changing these coefficients. In fact, these coefficients are not only important for BatchNorm, but also for other normalizations such as PixelNorm as shown in StyleGAN [2]. The authors acknowledge this in "Limitations and Future Work". Thus, this work does not really provide much insight on BatchNorm itself but on random features and conditioning, which makes the title, abstract and introduction misleading.
* It is also known that BatchNorm coefficients can produce sparsity [3]. I suggest the authors to update the text to account for this fact.
* Besides the main experiment of training BatchNorm only vs different alternatives, it would have been interesting if the authors had provided the results for alternative transformations. For instance, what would happen if only the PReLU parameters were trained? and if only Squeeze and Excitation blocks were trained [7]? Does the expressiveness of the affine transform change if using pre-activation or post-activation blocks [5]?
* Although the authors cite [8], they do not provide any insight if the "activation prunning" performed by $\gamma$ could be helping to find lottery tickets [9, 10]. It would be interesting if the authors added some result or new insight about that.
* Questions and Clarifications:
  * For the experiment where 2 random params per channel are trained, are these parameters "pixels" of convolutional filters? In this case, it may be normal that an affine transformation works better since the number of random parameters of the convolution will overwhelm the number of non-random ones. To be more fair, you could train a depth-wise separable convolution [6], and change an equivalent number of parameters as BatchNorm.
  * What happens with BatchNorm and its parameters when two parameters per channel are learned? Are they randomly initialized and frozen?
  * Does the standardization performed by BatchNorm boost the performance that can be obtained by the affine transformation?

Typos
--------
Page 3: "train[]"


[1] Perez, Ethan, et al. "Film: Visual reasoning with a general conditioning layer." arXiv preprint arXiv:1709.07871 (2017).

[2] Karras, Tero, Samuli Laine, and Timo Aila. "A style-based generator architecture for generative adversarial networks." Proceedings of the IEEE conference on computer vision and pattern recognition. 2019.

[3] Mehta, Dushyant, Kwang In Kim, and Christian Theobalt. "On implicit filter level sparsity in convolutional neural networks." Proceedings of the IEEE Conference on Computer Vision and Pattern Recognition. 2019.

[4] He, Kaiming, et al. "Delving deep into rectifiers: Surpassing human-level performance on imagenet classification." Proceedings of the IEEE international conference on computer vision. 2015.

[5] He, Kaiming, et al. "Identity mappings in deep residual networks." European conference on computer vision. Springer, Cham, 2016.

[6] Chollet, François. "Xception: Deep learning with depthwise separable convolutions." Proceedings of the IEEE conference on computer vision and pattern recognition. 2017.

[7] Hu, Jie, Li Shen, and Gang Sun. "Squeeze-and-excitation networks." Proceedings of the IEEE conference on computer vision and pattern recognition. 2018.

[8] Zhou et al. (2019) and Ramanujan et al. (2019) in the submission.

[9] Frankle, Jonathan, and Michael Carbin. "The lottery ticket hypothesis: Finding sparse, trainable neural networks." arXiv preprint arXiv:1803.03635 (2018).

[10] Malach, Eran, et al. "Proving the Lottery Ticket Hypothesis: Pruning is All You Need." arXiv preprint arXiv:2002.00585 (2020).


-----------------------
After Rebuttal
============
My main concern was that it is already known that BatchNorm has a great expressive power, and thus the authors should have gone broader and deeper in their search for valuable insights to explain what makes BatchNorm so special. I proposed different ways to do that, such as comparing with other parametric transformations like squeeze-and-excitation, or trying to understand what is the role of normalization. My concerns were shared with Reviewer 4, who proposed an interesting experiment that involves comparing multiple normalization functions such as LayerNorm.

The authors have agreed with some of these points, and updated the text to reflect the discussion. However, they believe all these suggestions belong to future work and, although the form of the paper has changed, the content is still the same.

Overall, I think this work is interesting and I think studies like this are necessary (although more in depth). Thus,  I have raised my score to borderline.

---

> ### Author Response · Authors · 2020-11-25
> **Author Response (Part 2)**
>
> _Although the authors cite [8], they do not provide any insight if the "activation prunning" performed by γ could be helping to find lottery tickets [9, 10]. It would be interesting if the authors added some result or new insight about that._
>
> We emphatically stress that our work is not directly related to lottery tickets. Work on the lottery ticket hypothesis focuses on pruning individual weights, whereas the sparsity we see here is in the form of pruning entire channels. These two forms of sparsity behave differently: another paper shows that methods that prune entire convolutional channels (as we do here) do not behave according to the claims in the lottery ticket work; unlike lottery tickets, it is possible to reinitialize the networks and train to the same accuracy [a]. As such, we chose not to compare our work to work on the lottery ticket hypothesis, since the sparsity we see is at a different granularity that is known to have different properties.
>
> The reason we compare to [8] is that both of those works involve pruning networks whose weights are frozen at their initializations, which - in that sense - is similar to our setting. However, both of the papers in [8] again involve pruning individual weights, while we look at pruning entire convolutional channels. As such, those papers are not directly comparable to our work beyond the fact that, as we state, they share our observation that “the raw material present at random initialization is sufficient to create performant networks” (Section 6).
>
> ---
> _For the experiment where 2 random params per channel are trained, are these parameters "pixels" of convolutional filters? In this case, it may be normal that an affine transformation works better since the number of random parameters of the convolution will overwhelm the number of non-random ones. To be more fair, you could train a depth-wise separable convolution [6], and change an equivalent number of parameters as BatchNorm._
>
> We completely agree with this intuitive explanation; it is the same one that we pose at the end of Section 4 to explain these results. Our goal in the experiment was to determine (1) whether  any subset of parameters similarly distributed throughout the network could reach this accuracy or (2) whether BatchNorm parameters were, parameter-for-parameter, able to reach higher accuracy. As such, we think this experiment was quite “fair.” However, we agree that a depth-wise separable convolution provides an excellent way to conduct this experiment in a context where each convolutional parameter will carry more weight, and we will look into adding MobileNets to the next revision of our paper (we weren’t able to do so during the discussion period).
>
> ---
> _What happens with BatchNorm and its parameters when two parameters per channel are learned? Are they randomly initialized and frozen?_
>
> They are disabled entirely. We fix all gamma parameters to 1 and all beta parameters to 0. This is to avoid a bad BatchNorm initialization interfering with the convolutions.
>
> ---
> _Does the standardization performed by BatchNorm boost the performance that can be obtained by the affine transformation?_
>
> We agree that this is an interesting question for future work. To disambiguate between the role of the affine parameters and the role of normalization (or to understand whether the two work best when combined), we would like to run the same experiment in contexts that do not need BatchNorm. For example, adding gamma and beta parameters to each channel in a convolutional network that does not have normalization or running our experiment on a ResNet trained with FixUp initialization [b] (an initialization scheme that does not require BatchNorm). We could then run these same experiments with normalization and compare the results. This experiment is beyond the scope of our results, which focus on BatchNorm in particular, but we believe this is an interesting direction for future work.
>
> [a] Zhuang Liu et al. Rethinking the Value of Network Pruning. ICLR 2019.
>
> [b] Hongyi Zhang et al. FixUp initialization: Residual Learning Without Normalization. ICLR 2019.

---

> ### Author Response · Authors · 2020-11-25
> **Author Response (Part 1)**
>
> We thank the reviewer for their detailed feedback. We have responded to all comments in-line below.
>
> ---
>
> _It is known that BatchNorm coefficients have a high expressive power. For instance, FILM [1] layers modulate neural networks by changing these coefficients. In fact, these coefficients are not only important for BatchNorm, but also for other normalizations such as PixelNorm as shown in StyleGAN [2]. The authors acknowledge this in "Limitations and Future Work". Thus, this work does not really provide much insight on BatchNorm itself but on random features and conditioning, which makes the title, abstract and introduction misleading._
>
> We agree that FILM [1] is another example of the expressive power of BatchNorm, and we have added it to our related work section. However, we disagree that our work “does not provide much insight on BatchNorm itself.” We demonstrate that gamma and beta have substantial expressive power on their own, even when they are not manipulating trained features as they do for FILM. We also demonstrate that, even in the context of training the network normally, gamma is smaller in the wider/deeper networks (Figure 5) where its presence has a meaningful impact on the overall accuracy of the network (Figure 1). This is in addition to the many observations we make about random features via our main experiment of training only BatchNorm.
>
> We have revised our abstract and introduction to better scope our work in the context of the literature. As you suggest, we have framed our contribution as a way to understand per-feature affine parameters; freezing all other weights at their random initializations makes it possible to distinguish the contribution of these affine parameters from the trained features that they typically transform, and BatchNorm is a convenient vantage point from which to study these parameters.
>
> Finally, we strongly disagree with the suggestion that our title, abstract, and introduction were  “misleading.” To the contrary, our subtitle is “on the expressive power of random features in CNNs,” a phenomenon we study by “Training BatchNorm and Only BatchNorm” - we believe this is a very precise description of our paper.
>
> ---
>
> _It is also known that BatchNorm coefficients can produce sparsity [3]. I suggest the authors to update the text to account for this fact._
>
> Although [3] also studies feature sparsity in convolutional networks (measured by both activation and the value of $\gamma$), their work is qualitatively different than ours in that it focuses on feature sparsity during the normal course of training rather than training only BatchNorm. While we do observe a small amount of feature sparsity during normal training (blue line in Figure 7), our claim is specific to feature sparsity when training only BatchNorm, a setting which Mehta et al. do not examine.
>
> In fact, even when training normally, we observe much lower levels of per-feature sparsity than Mehta et al. do (almost always less than 10% in our case vs. 50% in Mehta et al., despite the fact that our threshold for $\gamma$ is an order of magnitude higher than that of Mehta et al.). This may indicate that the behaviors Mehta et al. observe in their “BasicNet” setting (a 7-layer ConvNet for CIFAR) and VGG setting (it appears they use the ImageNet version of VGG, 500x larger than ResNet-20 and wildly overparameterized for CIFAR-10) may not hold in general.
>
>
> We have updated Section 5 to discuss the connection to Luo et al (third to last paragraph in Section 5).
>
> ---
> _Besides the main experiment of training BatchNorm only vs different alternatives, it would have been interesting if the authors had provided the results for alternative transformations. For instance, what would happen if only the PReLU parameters were trained? and if only Squeeze and Excitation blocks were trained [7]? Does the expressiveness of the affine transform change if using pre-activation or post-activation blocks [5]?_
>
> We agree that these are great questions, similar to your suggestion of looking at the parameters in depth-wise separable convolutions. In a similar vein, we are also interested in looking at other forms of normalization as Reviewer 4 mentioned. Although these ideas are outside the scope of the current paper (and we don’t think they qualify as a weakness), we believe they are exciting directions for future work.

---

### Official Review · AnonReviewer3 · 2020-10-25
**Review of Paper 310**

**Rating:** 6
**Confidence:** 4

**Review:**

This paper investigates the role of shit and bias parameters in BatchNorm. The authors find that by training BatchNorm parameters with other parameters fixed, the networks can already achieve a good performance. They also show a connection between BatchNorm parameters and disabling features.

Pros:
- The topic is interesting. Understanding the important role of gamma and beta offers great help to many relevant tasks since BatchNorm has been widely used in various networks.
- The paper is well written and organized.
- Extensive experiments validate the proposed findings.

Cons:
- The paper is more like a technical report. Although it draws many interesting conclusions about the BatchNorm parameters, the authors do not provide how these findings contribute to the community. I expect that the authors can show some promising results on applications. If not, some theoretical analyses may need to be provided.
- Some observations are not novel. For example, Mehta et al. [1] also found the connection between gamma and sparsity. The role of gamma has also been proved by Luo et al. [2]

In conclusion, this paper provides some insights into understanding the role of BatchNorm parameters. However, some observations are not novel and it is unclear how these findings help the community, which prevents this work to be accepted.

[1] On implicit filter level sparsity in convolutional neural networks. CVPR, 2019.

[2] Towards understanding regularization in Batch Normalization. ICLR, 2019.


****After rebuttal

I would appreciate the authors' detailed response to my concerns. It clearly clarifies their contributions compared with the previous work. Overall, the paper conducts comprehensive experiments and shows the expressive power of BN parameters. In particular, an interesting observation about the random feature and affine parameters is made. In the rebuttal, the author also gives some implications of their findings. I would like to increase my rating to 6.

---

> ### Author Response · Authors · 2020-11-25
> **Author Response (Part 2)**
>
>
> _Some observations are not novel...The role of gamma has also been proved by Luo et al. [7]._
>
> While Luo et al [7] study the gamma parameter in BatchNorm, we do not see why this affects the novelty of our work. Luo et al. show theoretically that using BatchNorm with weight decay on a single neuron leads to (among other behaviors) a phenomenon called “gamma decay,” where the gamma term is regularized in a data-dependent fashion.
>
> Luo et al.’s observation may indeed provide insight into why training only BatchNorm leads to per-parameter sparsity: by freezing all other parameters at their initial values, it is possible that our networks more closely adhere to Luo et al’s assumptions, and gamma decay may somehow encourage sparsity (although it is data-dependent L2 regularization rather than sparsity-inducing L1 regularization).
>
> Finally, our analysis is focused on understanding the behavior of gamma and beta in large-scale, practical settings. Although Luo et al. do include experiments, these experiments are intended to (a) compare BatchNorm with an explicit realization of PopulationNorm/Gamma decay and (b) to argue that there is a regularization effect on gamma (not that gamma leads to sparsity).
>
> In sum, the most important connection between Luo et al and our work is that we both study BatchNorm and analyze the gamma parameter--we hardly think that studying a similar problem from an entirely different perspective means that our work is not novel.
>
> We have updated Section 5 to discuss the connection to Luo et al (second to last paragraph in Section 5).
>
> [1] Identifying and Understanding Deep Learning Phenomena, ICML 2019 workshop, http://deep-phenomena.org/
>
> [2] Science meets Engineering of Deep Learning, NeurIPS 2019 workshop, https://sites.google.com/view/sedl-neurips-2019/main
>
> [3] Ali Rahimi, Test of Time award talk, NeurIPS 2017, https://www.youtube.com/watch?v=Qi1Yry33TQE
>
> [4] Amir Rosenfeld & John K. tsotsos. Intriguing Properties of Randomly Weighted Networks: Generalizing While Learning Next to Nothing. ArXiv.
>
> [5] Pramod Kaushik Mudrarkarta et al. K for the Price of 1: Parameter-Efficient Multi-Task and Transfer Learning. ICLR 2019.
>
> [6] Mehta et al. On implicit filter level sparsity in convolutional neural networks. CVPR 2019.
>
> [7] Luo et al. Towards understanding regularization in Batch Normalization. ICLR, 2019.

---

> ### Author Response · Authors · 2020-11-25
> **Author Response (Part 1)**
>
> We thank the reviewer for the detailed feedback. We have addressed the reviewer comments in-line below.
>
> ---
>
> _The paper is more like a technical report. Although it draws many interesting conclusions about the BatchNorm parameters, the authors do not provide how these findings contribute to the community. I expect that the authors can show some promising results on applications. If not, some theoretical analyses may need to be provided._
>
> _It is unclear how these findings help the community_
>
> While the reviewer is correct that our work does not propose any new applications, we would like to emphasize this was not the goal of this work. Rather, our aim was to rigorously study the affine parameters in batch normalization as they behave in practice. These parameters are ubiquitous in modern networks for computer vision, and the community still has limited empirical understanding of the practical role or expressive power of these kinds of learned affine transformations of features.
>
> While novel applications are of course important, careful empirical work aiming to better understand our existing techniques are equally important. This style of work is increasingly being recognized and sought by the community, as evidenced by several recent workshops and talks emphasizing rigorous empirical studies. BatchNorm remains particularly contentious, and the primary agreement in the community is that the initial motivation is not the source of its effectiveness. Therefore, novel observations about BatchNorm are highly desirable.
>
> And, while theory work is important, our current theory is not adequate to predict or describe many important phenomena that take place in deep neural networks at scale. It is not obvious what a theoretical component of this work would provide. One possibility would be to use theory to predict what initialization distributions optimize performance, but this is certainly beyond the scope of our work.
>
> In our view, the best way to understand these phenomena as they occur in practice is to investigate them empirically as they occur in practice. The distinctive quality of our paper is the level of depth and rigor with which we analyze gamma and beta when (1) training only BatchNorm and (2) when training all parameters. Although we are not the first to run an experiment where we train only BatchNorm [4, 5], we make many novel observations (e.g., the effect of scale on accuracy, the particular importance of BatchNorm parameters vs. other parameters in the network, the representations that form by virtue of the sparsity that gamma induces) by virtue of our rigor.
>
> In summary, there are more ways to make important scientific contributions to deep learning than just “applications” and “theory” (to the contrary, the natural sciences depend on rigorous empirical research). We believe that our empirical approach to understanding deep learning is an important way to establish scientific knowledge and that it is inappropriate to dismiss all research of this kind as "technical reports" unworthy of publication.
>
> ---
>
> _Some observations are not novel. For example, Mehta et al. [6] also found a connection between gamma and sparsity._
>
> Although Mehta et al. also study feature sparsity in convolutional networks (measured by both activation and the value of $\gamma$), their work is qualitatively different than ours in that it focuses on feature sparsity during the normal course of training rather than training only BatchNorm. While we do observe a small amount of feature sparsity during normal training (blue line in Figure 7), our claim is specific to feature sparsity when training only BatchNorm, a setting which Mehta et al. do not examine.
>
> In fact, even when training normally, we observe much lower levels of per-feature than Mehta et al. do (almost always less than 5% in nearly all cases we consider vs. 50% in Mehta et al., despite the fact that our threshold for $\gamma$ is an order of magnitude higher than that of Mehta et al.). This may indicate that the behaviors Mehta et al. observe in their “BasicNet” setting (a 7-layer ConvNet for CIFAR) and VGG setting (it appears they use the ImageNet version of VGG, 500x larger than ResNet-20 and wildly overparameterized for CIFAR-10) may not hold in general.
>
> We have updated Section 5 to include these points of comparison and distinction (third to last paragraph).

---

### Official Review · AnonReviewer2 · 2020-10-29
**This is a thought-provoking experimental result that shows optimizing only BN parameters is very effective for deep neural networks**

**Rating:** 8
**Confidence:** 4

**Review:**

This paper studies the effect of training BN parameters on training deep neural networks. The conclusion is striking: learning only BN parameters is enough when increasing the depth of the network. Authors have done extensive experiments to understand the effect of increasing the depth and width of the network. To stress the important role of BN parameters, the same number of parameters are chosen randomly and trained. Yet, it is observed BN parameters can obtain far better accuracy. Furthermore, an interesting observation is conducted on the distribution of BN parameters: when training only these parameters, a sparsity pattern is observed on the optimal parameters. While learning all parameters does not reach such a sparse pattern for BN parameters. The sparsity pattern indicates that an efficient network only needs to have a particular ground-truth connection between different units and the choice of weights is not important. This shows that random features imposed by neurons can create a very interesting function class when they are connected in a proper way.

I have a question to understand the results betters: When authors are talking about optimizing only BN parameters, do they mean that they did not optimize parameters of the last linear layer in the network (the one that is connected to outputs).

I have one concern about the way that the key message delivered here: BN parameters are very important and needed to be trained. Yet, the effectiveness of these parameters is tied to the randomness of the weights. Let me explain more about this. Recent studies show that a BN network with random weights (according to the standard initialization scheme for neural networks) provides more distinguishing features compare to vanilla networks (see for example "Understanding batch normalization" and "Batch Normalization Provably Avoids Rank Collapse").  This may be the reason why training only BN parameter is sufficient as the interesting features that are created by random weights. This hypothesis can be checked with a simple experiment: initialize the weights with a non-zero mean distribution (or any other non-standard initialization) then see whether still the training only BN parameters is effective. My (academic) guess is that you can easily find an initialization scheme for weights which makes the training only BN parameters ineffective. This result will provide more intuition about the coupling of the weights and BN parameters.

* I have read the authors' response and also other reviewers. In my view, this paper provides novel insights into batch normalization.

---

> ### Author Response · Authors · 2020-11-25
> **Author Response**
>
> We thank the reviewer for the detailed feedback. We have answered the reviewer questions below.
>
> ---
>
> _When author are talking about optimizing only BN parameters, do they mean they did not optimize parameters of the last linear layer in the network (the one that is connected to outputs)._
>
> That is correct, except in experiments where we explicitly state that we train the output layer (e.g., in Figure 2).
>
> ---
>
> _The effectiveness of these parameters is tied to the randomness of the weights...This may be the reason why training only BN parameter is sufficient as the interesting features that are created by random weights. This hypothesis can be checked by a simple experiment: initialize the weights with a non-zero mean distribution (or any other non-standard initialization) then see whether still the training only BatchNorm parameters is effective._
>
> Thanks for bringing this up. In fact, in Appendix E.1, we describe several experiments we performed to this effect. We shared your intuition that other initialization distributions might affect the efficacy of training only BatchNorm, so we tried training running the same experiments with a uniformly distributed initialization (vs. the Gaussian distribution we used in the main body), an orthogonal initialization, and a binarized initialization where each weight was set to {-$\sigma$, $\sigma$}. Interestingly, none of these initializations affected our results, suggesting that the behaviors we observe are indeed general to many different ways of initializing the network (even the particularly restrictive binarized initialization). We will look into adding an initialization scheme you suggest (with a non-zero mean) as well.

---

### Decision · Program_Chairs · 2021-01-07
**Final Decision**

**Decision:**

Accept (Poster)

**Comment:**

The paper provides an astonishingly simple experiment: the parameters in the network are fixed, but only the parameters in the BatchNorm (taking less than 1% of the total number of parameters) are trained and also the last linear layer is trained.
 The resulting networks provide better accuracies than training a random subset of the network. Another part of this work is the study of the effect of $\beta$ and $\gamma$ when doing full training.

Pros: - All the reviewers agree this is an interesting and important observation.
          - Contribution is clear and paper is well-written
          - In future, better understanding of different parameters may

Cons: A concern has been raised by one of the reviewers that it is more like a technical report
           Some previous work which studies the effect of $\gamma$ was not mentioned.

I think, the most interesting part is training only $\beta$ and $\gamma$. It will provide a ground for theoretical investigations of the properties of deep neural network models, and maybe lead to more efficient training algorithms.